# Promiscuous interactions and protein disaggregases determine the material state of stress-inducible RNP granules

**Sonja Kroschwald, Shovamayee Maharana, Daniel Mateju, Liliana Malinovska, Elisabeth Nüske, Ina Poser, Doris Richter, Simon Alberti\***

Max Planck Institute of Molecular Cell Biology and Genetics, Dresden, Germany

**Abstract** RNA-protein (RNP) granules have been proposed to assemble by forming solid RNA/protein aggregates or through phase separation into a liquid RNA/protein phase. Which model describes RNP granules in living cells is still unclear. In this study, we analyze P bodies in budding yeast and find that they have liquid-like properties. Surprisingly, yeast stress granules adopt a different material state, which is reminiscent of solid protein aggregates and controlled by protein disaggregases. By using an assay to ectopically nucleate RNP granules, we further establish that RNP granule formation does not depend on amyloid-like aggregation but rather involves many promiscuous interactions. Finally, we show that stress granules have different properties in mammalian cells, where they show liquid-like behavior. Thus, we propose that the material state of RNP granules is flexible and that the solid state of yeast stress granules is an adaptation to extreme environments, made possible by the presence of a powerful disaggregation machine.

## Introduction

The organization of the intracellular space into compartments is fundamental to life; it allows cells to perform complex biochemical reactions in a confined and controlled manner. In addition to membrane-delimited organelles, such as the Golgi apparatus or the endoplasmic reticulum, the cytoplasm contains compartments that are not surrounded by membranes. These non-membrane-bound compartments are frequently linked to gene expression pathways and often contain substantial amounts of RNAs (*Hyman and Brangwynne, 2011*; *Weber and Brangwynne, 2012*; *Brangwynne, 2013*; *Hyman et al., 2014*). Examples in the nucleus include chromatin domains, the nucleolus, and Cajal bodies; examples in the cytoplasm are P bodies, stress granules, and germ granules. Although their functions are not always clear, it is assumed that they facilitate or suppress specific biochemical reactions.

Recent studies proposed that P granules—RNA-protein (RNP) granules in germ cells of *Caenorhabditis elegans*—form by liquid–liquid demixing or phase separation from the cytoplasm, so that two phases, a liquid droplet phase and the cytoplasm coexist (*Brangwynne et al., 2009*; *Lee et al., 2013*). Such demixed phases of proteins and RNAs may turn out to be a unifying principle of subcellular organization (*Weber and Brangwynne, 2012*; *Brangwynne, 2013*; *Hyman et al., 2014*). Indeed, recent findings show that liquid phase separation is not limited to P granules (*Brangwynne et al., 2011*; *Aggarwal et al., 2013*; *Feric and Brangwynne, 2013*; *Hubstenberger et al., 2013*; *Wippich et al., 2013*; *Banjade and Rosen, 2014*). Importantly, defined mixtures of RNAs and proteins can also phase separate in a cell-free system, driven by multivalent interactions (*Li et al., 2012*). This suggests that many cellular components may have an intrinsic ability to phase separate and assemble into structures with liquid-like properties.

Despite recent evidence for a liquid-like state, others have argued that RNP granules have more solid material characteristics, similar to protein aggregates (*Gilks et al., 2004*; *Vessey et al., 2006*;

**\*For correspondence:** alberti@mpi-cbg.de

**Competing interests:** The authors declare that no competing interests exist.

**eLife digest** Genes consist of long stretches of DNA that code for proteins. The DNA is first 'transcribed' to produce an RNA molecule, which is then translated into a protein. In most cells, RNA molecules are present within a structure called ribonucleoprotein (RNP for short) granules. These contain the protein machinery needed to transport, store, and break down RNAs.

P bodies and stress granules are two types of RNP granules found in all cells, from yeast to human. P bodies are present at all times, whereas stress granules assemble when a cell experiences stressful conditions, such as a lack of nutrients or high temperatures. Once the stress has been overcome, the stress granules are disassembled.

The precise details of how RNP granules assemble in cells remain poorly understood. One theory suggests that RNP granules form through a physical process called 'phase separation' in which RNA molecules and proteins above a certain critical concentration condense to form a liquid droplet. Other research has suggested that RNP granules arise when so-called prion-like proteins spontaneously clump together and start aggregating to form fibers. These granules would behave more like solids than liquids.

Kroschwald et al. have now analyzed how P bodies and stress granules form in yeast and human cells using a chemical compound that can distinguish between liquid-like and solid-like structures. The results revealed that P bodies and stress granules behave very differently in yeast cells. While P bodies are indeed liquid droplets, stress granules are more solid in nature and act like protein aggregates. So why is there a difference between the two? It is known from previous work that when cells are stressed, many proteins misfold and start aggregating. Kroschwald et al. found that the formation of stress granules coincides with the formation of aggregates, suggesting that stress granules themselves are a type of aggregate. Furthermore, stress granule formation does not seem to involve prion-like fibers, but rather prion-like proteins can easily interact with other proteins in a promiscuous way, thus promoting the seeding of stress granules and their growth.

Kroschwald et al. next studied human cells and observed that in these cells, both P bodies and stress granules were liquid droplets. These results together suggest that the physical properties and method of assembling P bodies and stress granules can vary from one organism to another. Future work will investigate whether the ability to form solid rather than liquid stress granules provides extra protection to yeast cells when they are stressed. It also remains to be tested whether and how stress granules convert into the pathological RNP aggregates that are often seen in neurodegenerative diseases.

*Decker et al., 2007*; *Si et al., 2010*; *Decker and Parker, 2012*; *Ramaswami et al., 2013*). This seems particularly likely for P bodies and stress granules, which are stress-inducible RNP granules containing non-translating RNAs and protein factors involved in translation repression or mRNA decay (*Anderson and Kedersha, 2009*; *Decker and Parker, 2012*). P bodies and stress granules form as a response to acute stress conditions, when a cell has to make arrangements to divert valuable resources to cellular survival, and their formation coincides with the formation of protein aggregates, which result from stress-induced protein misfolding.

Proteins contained in P bodies or stress granules are characteristically composed of two types of domains: RNA-binding domains (RBDs) and domains of low sequence complexity; the latter are also referred to as prion-like (*Gilks et al., 2004*; *Decker et al., 2007*; *Reijns et al., 2008*; *King et al., 2012*; *Malinovska et al., 2013*). This term is derived from the fact that they have a characteristic amino acid composition (mostly polar amino acids such as serine, glycine, asparagine, glutamine and tyrosine), which resembles that of yeast prions. Prion domains (PDs) and prion-like domains (PLDs) have little structure under normal conditions. However, they can undergo spontaneous conversions into an aggregated state, which is characterized by a cross-$\beta$ structure and referred to as amyloid (*Alberti et al., 2009*). Once formed, the amyloid state can act as a template for the incorporation of further proteins, but it can also be reversed through energy-expending protein disaggregation machines (*Doyle et al., 2013*).

Despite some recent progress, RNP granule assembly remains a poorly understood molecular process. In particular, it is not known how RBDs and PLDs cooperate to promote RNP granule

formation in living cells. Recent cell-free reconstitution experiments seem to confirm that PLDs in RNP granule components can assemble into amyloid-like fibers and undergo sol–gel or liquid–solid phase transitions (*Han et al., 2012*; *Kato et al., 2012*; *Kwon et al., 2013*, *2014*). However, PLDs can also cause protein-misfolding diseases, which are typically accompanied by solid RNP aggregates (*Li et al., 2013*; *Ramaswami et al., 2013*). These pathological RNP aggregates are similar to those observed in cell-free reconstitution experiments, raising important questions about the relationship of physiological and pathological RNP granules in living cells. More insight into this topic is also desirable in light of the fact that two different models have been proposed to explain RNP granule assembly, which make very different predictions about the material state of RNP granules (liquid vs solid) and their mode of assembly (phase separation vs aggregation).

In this study, we analyze P bodies and stress granules in budding yeast and mammalian cells. To our surprise, we find a high degree of versatility in RNP granule assembly. We show that physiological RNP granules can have different material properties and behave as liquid-like droplets or solid protein aggregates. We further establish a key role for RNA in RNP granule assembly and demonstrate that PLDs in RNA-binding proteins promote RNP granule formation in a manner that does not involve amyloid-like aggregation. Instead, these domains undergo promiscuous interactions, with other PLDs or with misfolded proteins. We further reveal a central role for ATP-driven disaggregases in maintaining the identity and integrity of RNP granules and propose that the presence of the Hsp104 disaggregase in yeast has enabled the evolution of a unique pathway for RNP granule formation, which resembles a typical protein aggregation reaction.

## Results

### PLDs can form two types of assemblies, only one of which is amyloid-like

To analyze the molecular mechanisms underlying the formation of stress-inducible RNP granules, we first focused on the role of prion-like proteins. These proteins have been implicated in RNP granule assembly (*Gilks et al., 2004*; *Vessey et al., 2006*; *Decker et al., 2007*; *Reijns et al., 2008*; *Si et al., 2010*; *Kato et al., 2012*), but they are also key determinants of protein aggregation (*Alberti et al., 2009*; *King et al., 2012*; *Malinovska et al., 2013*). Indeed, Decker and colleagues demonstrated in 2007 that the C-terminal domain of the protein Lsm4 is required for P-body formation in yeast. This domain is aggregation-prone and compositionally similar to yeast prion proteins (*Alberti et al., 2009*) (*Figure 1—figure supplement 1*). Consistent with this, the PD of the yeast prion Rnq1 could functionally replace the PLD of Lsm4, thus restoring P-body formation in certain genetic backgrounds (*Decker et al., 2007*). However, it remained undetermined whether the PD of Rnq1 adopted an amyloid-like conformation during P-body assembly or whether other properties of this domain were required.

To get insight into this question, we analyzed the PD of Rnq1 in yeast cells. First, we studied its subcellular localization in yeast carrying the background prion [*PIN+*], a factor that promotes the conversion of the Rnq1 protein and other yeast prion proteins into an amyloid state. In these cells, GFP (green fluorescent protein)-tagged Rnq1PD assembled into punctate structures (*Figure 1A*). To determine whether Rnq1PD could also aggregate in the absence of a co-inducing prion, we expressed it in a [*pin−*] background. The resulting cells also displayed a punctate fluorescence signal (*Figure 1A*). However, the pattern of aggregation was different, because not all the signal was concentrated in foci (*Figure 1—figure supplement 2*). Next, we analyzed these cells by semi-denaturing detergent-agarose gel electrophoresis (SDD-AGE); this method can separate SDS (sodium dodecyl sulfate) -soluble from SDS-insoluble fractions, SDS insolubility being a hallmark of amyloid polymers (*Alberti et al., 2010*). Using this technique, we found that Rnq1PD formed SDS-resistant polymers in [*PIN+*] cells (*Figure 1B*). The Rnq1PD structures in [*PIN+*] cells could also be stained with the amyloid-specific dye Thioflavin T (ThT) (*Figure 1C*). In contrast, Rnq1PD expressed in [*pin−*] cells did not form SDS-resistant aggregates (*Figure 1B*) and it could not be stained with ThT (*Figure 1C*). Thus, we conclude that Rnq1PD can form two types of assemblies in yeast cells: amyloid-like aggregates and non-amyloid assemblies.

### Hexanediol can disrupt non-amyloid assemblies formed by PLDs

Our next goal was to develop a tool that could differentiate between these two assembled states of Rnq1PD. To do this, we made use of the chemical 1,6-hexanediol, an aliphatic alcohol that

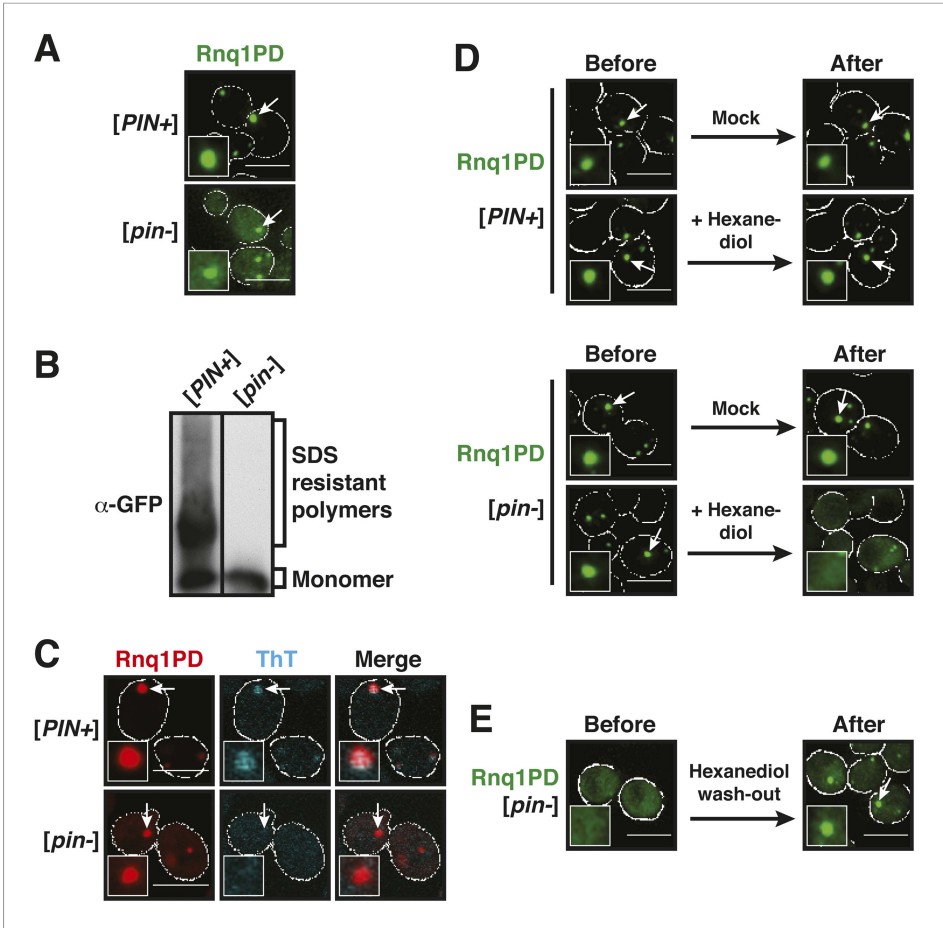

Figure 1. Prion-like domains can access two distinct aggregated states, only one of which is amyloid-like. (A) Fluorescence microscopy of yeast cells expressing sfGFP-tagged Rnq1PD in [PIN+] and [pin−] cells. White lines indicate the cell boundaries. Scale bars: 5 μm. Also see related Figure 1—figure supplements 1–3. (B) Semi-denaturing detergent-agarose gel electrophoresis (SDD-AGE) of [PIN+] and [pin−] cells containing a plasmid for expression of Rnq1PD-sfGFP. SDS-resistant amyloid polymers show slower migration in comparison to SDS-soluble monomers. Proteins were detected by immunoblotting with a GFP-specific antibody. (C) Thioflavin T (ThT) staining of [PIN+] and [pin−] cells expressing Rnq1PD-mCherry from a plasmid. Note that only amyloid-like assemblies can be stained with ThT. (D) 1,6-hexanediol treatment specifically disrupts non-amyloid Rnq1PD assemblies and not amyloids. Fluorescence time-lapse microscopy of [PIN+] (top panel) and [pin−] cells (bottom panel) expressing Rnq1PD-sfGFP. Time points are before treatment (Before) and 38 min after treatment with 10% 1,6-hexanediol (After). In the control condition (Mock) only media was added. Cells were permeabilized with 10 μg/ml digitonin. See corresponding Video 1. (E) Cells expressing Rnq1PD-sfGFP were treated with 10% 1,6-hexanediol and digitonin for 1 hr to dissolve non-amyloid Rnq1PD assemblies (Before). Hexanediol was washed out and replaced with normal growth media (After), and the cells were observed with fluorescence microscopy. Also see corresponding Video 2.

The following figure supplements are available for figure 1:

Figure supplement 1. Lsm4 has a prion-like C-terminal domain (underlined) that is enriched for asparagines (N) and glutamines (Q) and contains hydrophobic residues (L, V, I, M, F).

Figure supplement 2. Rnq1PD shows a different aggregation pattern in [pin−] and [PIN+] cells.

Figure supplement 3. The amino acid sequence of the prion domain of Rnq1 resembles that of FG repeat-containing low-complexity domains of nucleoporins (hydrophobic aromatic residues in an asparagine- and glutamine-rich polar sequence background).

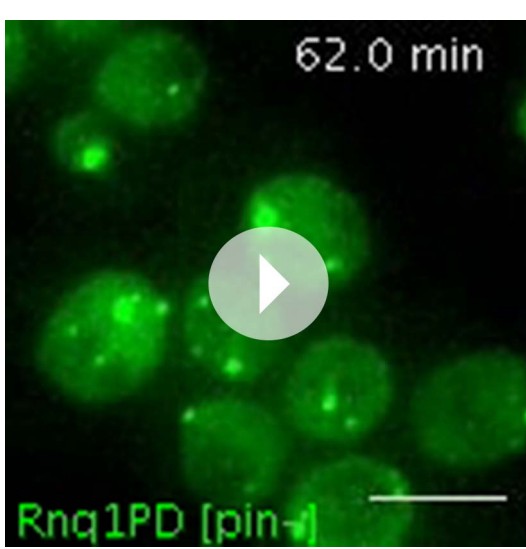

**Video 1.** Hexanediol treatment specifically disrupts non-amyloid Rnq1PD assemblies and not amyloids. Fluorescence time-lapse microscopy of [*PIN+*] and [*pin−*] cells expressing Rnq1PD-sfGFP. All cells were treated with 10 μg/ml digitonin and, where indicated, with 10% 1,6-hexanediol. In the control condition only media was added. Related to *Figure 1*.

has previously been shown to perturb nucleoporin-mediated transport across the nuclear pore (*Ribbeck and Gorlich, 2002*; *Patel et al., 2007*). Many nucleoporins contain domains of low sequence complexity, which are compositionally similar to yeast PDs and form a sieve-like matrix that enables the selective passage of cargo complexes (*Frey et al., 2006*; *Frey and Gorlich, 2007*; *Hulsmann et al., 2012*). Importantly, sieve formation involves weak hydrophobic interactions between phenylalanine–glycine repeats that are embedded in the PLD. When these interactions are perturbed by 1,6-hexanediol, nucleocytoplasmic transport ceases (*Ribbeck and Gorlich, 2002*; *Patel et al., 2007*).

The amino acid composition of the PD of Rnq1 is similar to that of nucleoporins (*Figure 1—figure supplement 3*), suggesting that Rnq1PD aggregation may likewise be affected by hexanediol. Hexanediol indeed triggered the dissolution of non-amyloid Rnq1PD assemblies, whereas the amyloid form remained unaffected (*Figure 1D*, *Video 1*). Importantly, this process was reversible (*Figure 1E*, *Video 2*). This suggests that hexanediol can differentiate between these two types of assemblies and may thus be a powerful tool to interfere with the formation of structures that depend on weak interactions between sticky PLDs.

## The prion-like protein Lsm4 does not adopt an amyloid-like conformation in P bodies

The PLD of yeast Lsm4 is required for the assembly of P bodies (*Decker et al., 2007*). However, the PLD of Lsm4 has not only been implicated in RNP granule formation, it also assembles into an amyloid-like state (*Alberti et al., 2009*). Because the induction of an amyloid state is a concentration-dependent nucleation process, the conformational conversion of prion-like proteins into amyloid can be induced in [*PIN+*] yeast cells by raising their concentration. Thus, by expressing additional Lsm4 from a plasmid in the [*PIN+*] background, we could genetically drive endogenous Lsm4 into an amyloid state. In this state, endogenous Lsm4 formed one to a few fluorescent puncta in the cytoplasm (*Figure 2A*, upper panel). Lsm4 was also present in puncta in uninduced cells, but they were smaller (*Figure 2A*, left lower panel). Previous studies identified these puncta as P bodies (*Decker et al., 2007*). Because P-body formation is strongly enhanced by stress, we next analyzed the localization of Lsm4 after glucose depletion. Now, Lsm4 was present in a few large puncta per cell (*Figure 2A*, right lower panel), and this stress-induced localization pattern was reminiscent of the pattern formed by Lsm4 in the amyloid conformation. We further observed that other P-body components such as Edc3 and Dcp2 co-localized with amyloid-like Lsm4 structures (*Figure 2B*). One potential explanation for this is that amyloid formation by Lsm4 occurs during the formation of P bodies.

To investigate this possibility, we analyzed Lsm4 in the P-body state. First, we generated lysates from stressed cells and subjected them to

**Video 2.** Reformation of non-amyloid Rnq1PD assemblies after hexanediol removal. Cells expressing Rnq1PD-sfGFP were treated with 10% 1,6-hexanediol and 10 μg/ml digitonin for 1 hr to dissolve non-amyloid Rnq1PD assemblies. Hexanediol was washed out and replaced with normal growth media and the cells were observed by fluorescence time-lapse microscopy. Related to *Figure 1*.

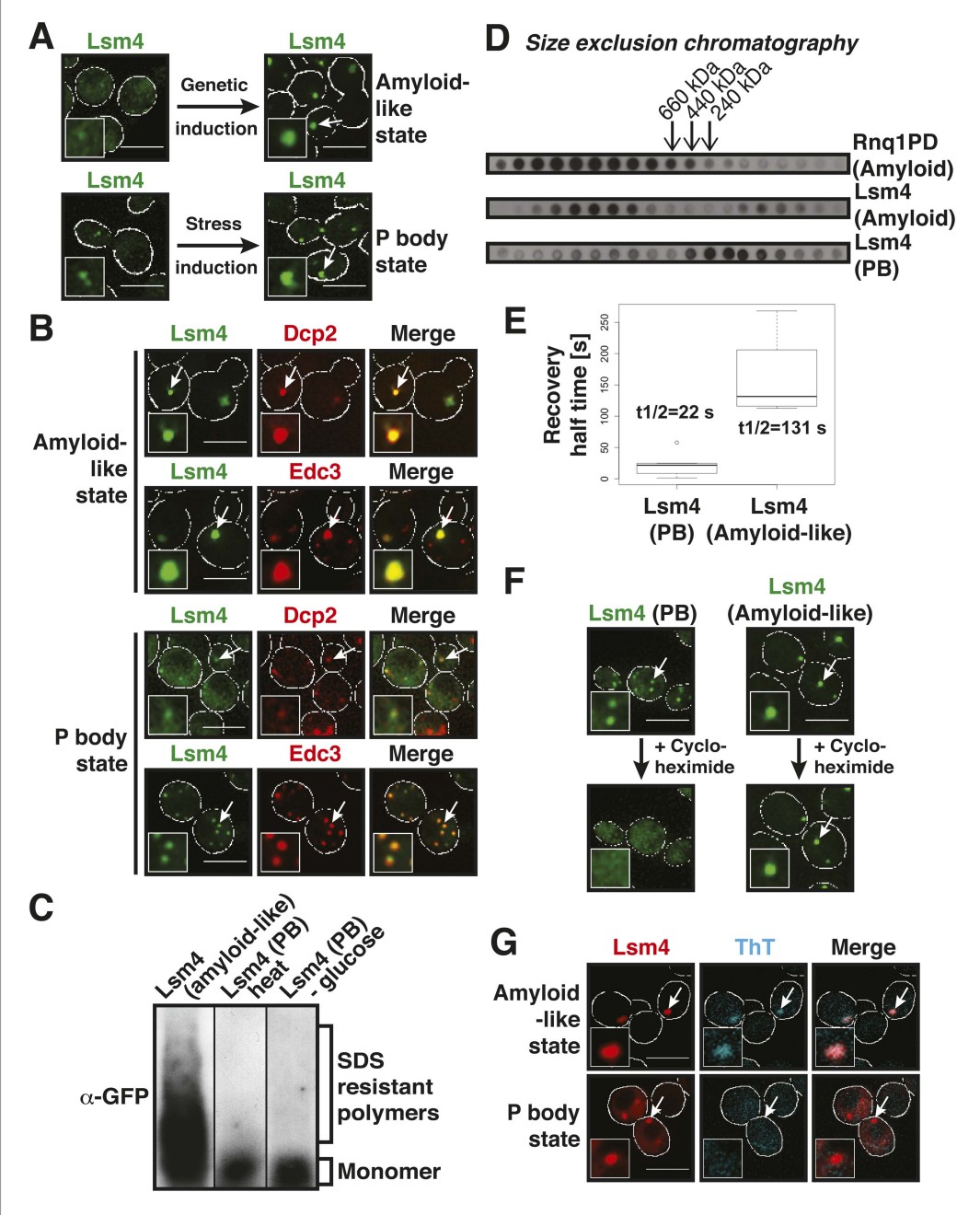

**Figure 2**. The prion-like protein Lsm4 does not adopt an amyloid-like conformation in P bodies. (**A**) Genetically induced and stress-induced Lsm4 assemblies are morphologically similar. Endogenous GFP-tagged Lsm4 in the amyloid-like state and PB state was investigated by fluorescence microscopy. PB formation was induced by glucose starvation for 1 hr (Stress induction). Genetic induction of Lsm4-GFP assembly into an amyloid-like state was through overexpression of unlabeled Lsm4. White lines indicate the cell boundaries. Scale bars: 5 μm. (**B**) Fluorescence microscopy of Lsm4-GFP cells co-expressing the mCherry-tagged PB proteins Dcp2 or Edc3. Lsm4 assembly was induced genetically (amyloid-like state) or through heat stress at 46°C for 10 min (P-body state). Note that both types of assemblies show co-localization with Dcp2 and Edc3. (**C**) Lsm4 in P bodies does not adopt an amyloid-like conformation. Comparative SDD-AGE analysis of Lsm4-GFP in the amyloid-like and PB state. P bodies were induced through heat stress or glucose depletion. (**D**) The Lsm4 RNP complex in P bodies has a different size than the amyloid-like Lsm4 complex. Size exclusion chromatography of Lsm4-GFP in the amyloid-like state and PB state. Cells expressing Rnq1PD from a plasmid were used as a control for amyloid. Molecular size standards were: thyroglobulin (660 kDa), ferritin (440 kDa), and catalase (240 kDa). Cells were stressed in glucose-deficient medium to

*Figure 2. continued on next page*

*Figure 2. Continued*

induce PBs. (**E**) Fluorescence recovery after photobleaching analysis of Lsm4 in the P body state (induced by glucose depletion) or amyloid-like state. Dendra2-tagged Lsm4 was expressed from a plasmid. The median half-recovery times were 22 s for the PB state (n = 6) and 131 s for the amyloid-like state (n = 4). Using the unpaired Student's *t*-test, the values are significantly different with p = 0.0018. Also see related *Figure 2—figure supplement 1*.
(**F**) Lsm4-containing P bodies are RNA-dependent, whereas amyloid-like Lsm4 assemblies are not. Cells containing Lsm4-GFP in the PB state (induced by glucose depletion) or amyloid-like state were treated with 100 µg/ml cycloheximide and observed by fluorescence time-lapse microscopy after 14 min. See corresponding *Video 3*. (**G**) P bodies are not stainable with the amyloid-specific dye ThT. Fluorescence microscopy of ThT-stained cells expressing mCherry-tagged Lsm4. Lsm4 assembly was induced genetically (amyloid-like state) or through glucose depletion (P-body state). Also see related *Figure 2—figure supplement 2*.

The following figure supplements are available for figure 2:

**Figure supplement 1**. Lsm4 in P bodies shows faster turnover than Lsm4 in amyloid-like assemblies.

**Figure supplement 2**. P-body proteins do not enter into an amyloid-like state upon stress.

SDD-AGE analysis. As can be seen in *Figure 2C*, Lsm4 did not become resistant to SDS in response to glucose depletion or robust heat stress. In contrast, SDS-resistant polymers were readily detected, when Lsm4 was genetically driven into an amyloid conformation (*Figure 2C*). We then prepared lysates from stressed cells in mild detergent (in the absence of SDS) and subjected them to size exclusion chromatography (SEC). Lsm4 in the amyloid state formed very large complexes, whereas Lsm4 complexes in stressed cells were much smaller (*Figure 2D*). As a next step, we investigated the dynamic behavior of Lsm4 in P bodies. To do this, we tagged Lsm4 with the fluorophore Dendra2. The resulting yeast strain was stressed by removal of glucose, and P body-localized Lsm4-Dendra2 was photo-converted from green to red. The recovery of the green fluorescence in the P bodies was then followed over time. As can be seen in *Figure 2E* and *Figure 2—figure supplement 1*, the turnover of Lsm4 in P bodies was rapid and in the same range as previously reported for mammalian P-body components (*Andrei et al., 2005*; *Kedersha et al., 2005*; *Aizer et al., 2008*). However, the turnover rate of Lsm4 in the amyloid conformation was much slower (*Figure 2E* and *Figure 2—figure supplement 1*). Next, we tested whether Lsm4 assemblies are dependent on RNA for their formation, as demonstrated previously for other RNP granules (*Andrei et al., 2005*; *Teixeira et al., 2005*). Indeed, stress-induced Lsm4-labeled P bodies disassembled in the presence of the translation inhibitor cycloheximide, an inhibitor that traps RNAs on polysomes and thereby depletes P bodies of RNA substrates (*Figure 2F*, *Video 3*). In contrast, Lsm4 in the amyloid state was unaffected by cycloheximide, suggesting that the formation of this structure is independent of RNA.

Collectively, these findings indicated that the Lsm4 conformation in stressed cells is fundamentally different from the experimentally induced amyloid conformation. To more generally test whether P-body formation depends on amyloid-like aggregation, we treated P body-containing yeast cells with the amyloid-specific dye ThT. As shown in *Figure 2G*, Lsm4-containing P bodies were ThT negative, whereas amyloid-like assemblies of Lsm4 were readily identified by ThT. Moreover, other P-body proteins with PLDs, such as Edc3 and Dcp2, did not become SDS-resistant in stressed cells (*Figure 2—figure supplement 2*), suggesting that they do not enter into amyloid-like states. Thus, we conclude that amyloid-like conformational conversions are not required for the formation of P bodies in yeast.

## Yeast P-bodies are liquid-like droplets and not aggregates

Our data so far suggest that P-body formation may not be dependent on amyloid-like conformational changes. They further imply that yeast P-bodies are dynamic RNA-dependent structures. Recent findings in *C. elegans* embryos indicate that P granules—germ line RNP granules related to P bodies—have liquid-like properties and form by demixing from the cytoplasm (*Brangwynne et al., 2009*; *Lee et al., 2013*). Thus, we hypothesized that the assembly of yeast P-bodies may be governed by the same physical principle.

The morphology of a structure or its ability to fuse can provide important hints about its material state (*Hyman et al., 2014*). Indeed, fluorescence microscopy of yeast P-bodies revealed a smooth spherical surface, in agreement with a liquid-droplet state (*Figure 3A*). Moreover, P bodies underwent

**Video 3.** P-body formation requires RNA. Glucose starved cells containing Lsm4-GFP in the PB state or amyloid-like state ('Prion') were treated with 100 μg/ml cycloheximide and observed by fluorescence time-lapse microscopy. Related to *Figure 2*.

frequent fusion events (*Figure 3A,B*). Upon fusion, the newly formed body rapidly relaxed into a spherical shape (*Figure 3A*). Such fast relaxation times indicate that the viscosity of P bodies is relatively low, in agreement with a dynamic liquid-like state.

A liquid state requires that the molecular interactions are weak and permanently changing (*Hyman et al., 2014*). Solid states instead are based on tight interactions, which are largely invariant over time. We reasoned that hexanediol could be a useful tool to differentiate between liquid-like and solid-like states, because of its ability to interfere with weak hydrophobic interactions. In fact, hexanediol has been used previously to assess the liquid-like nature of germ granules in *C. elegans* (*Updike et al., 2011*). Indeed, when we added hexanediol to yeast cells, it dissolved P bodies (*Figure 3C*, *Video 4*). Importantly, this effect was rapid (*Figure 3—figure supplement 1*) and reversible, as P bodies reformed after hexanediol washout (*Figure 3D*, *Video 5*). This suggests that P bodies—similar to germ granules in *C. elegans*—rely on weak interactions for their formation. Thus, we conclude that yeast P-bodies resemble liquid droplets with physicochemical properties unlike those of solid amyloid-like aggregates.

## Yeast stress granules have different material properties than P bodies

Stress granules are related to P bodies; they are induced by stress and function as storage depots for mRNAs (*Anderson and Kedersha, 2008*; *Decker and Parker, 2012*). We therefore wondered whether yeast stress granules show similar liquid-like behavior as P bodies. To investigate this, we tested the effect of hexanediol on stress granules. Remarkably, hexanediol did not affect stress granule integrity, even when applied for extended times or in the presence of digitonin to facilitate hexanediol entry into the cells (*Figure 4A*, *Figures 4—figure supplement 1*, *Video 6*). Thus, we conclude that stress granules are distinct from P bodies and may instead have a more solid character.

Like the constituent proteins of P bodies, many stress granule proteins contain PLDs. To investigate whether these proteins undergo amyloid-like conformational conversions, we initially focused on one stress granule protein: Nrp1. Nrp1 is a prototypical stress granule protein in that it contains a RNA-binding domain (RRM - RNA recognition motif) and a PLD (*Buchan et al., 2008*). However, Nrp1 can also be converted into an amyloid-like state (*Alberti et al., 2009*). Moreover, it is one of the proteins capable of binding to b-isox (*Kwon et al., 2013*), a chemical that specifically binds to low-complexity domains, which have the ability to undergo amyloid-like conformational conversions (*Kato et al., 2012*).

To investigate whether amyloid-like aggregation of Nrp1 is involved in stress granule assembly, we first converted endogenous Nrp1 into an amyloid state, using a genetic approach (see 'Materials and methods' for details). In the resulting cells, Nrp1 localized to one or a few bright cytoplasmic foci (*Figure 4B*, top panel). The same localization pattern was observed in stressed cells (*Figure 4B*, lower panel). Both structures co-localized with other stress granule proteins such as Pab1 (*Figure 4C*), which is consistent with the possibility that genetically induced amyloid-like Nrp1 structures are genuine stress granules. Thus, we next investigated whether stress triggers an amyloid-like conversion in Nrp1 using SDD-AGE. However, Nrp1 did not become SDS-resistant in response to stress (*Figure 4D*). Nrp1 complexes also had a very different molecular size in stressed cells (*Figure 4—figure supplement 2*). In addition, we found that the turnover rate of amyloid-like Nrp1 was much slower (*Figure 4E*, *Figure 4—figure supplement 3*). As a next step, we investigated four additional stress granule proteins with PLDs using SDD-AGE. However, none of these proteins formed amyloid-like structures in stressed cells (*Figure 4—figure supplement 4*). Furthermore, stress-induced Nrp1 assemblies were ThT negative, whereas genetically induced Nrp1 assemblies were stainable with ThT (*Figure 4F*). Thus, we conclude that stress granules are more solid-like but like P bodies do not transition into amyloid-like states upon stress.

## Yeast stress granules resemble amorphous protein aggregates

Why do stress granules adopt a different material state than P bodies? A previous study showed that in heat-stressed cells, the formation of stress granules coincides with the formation of protein

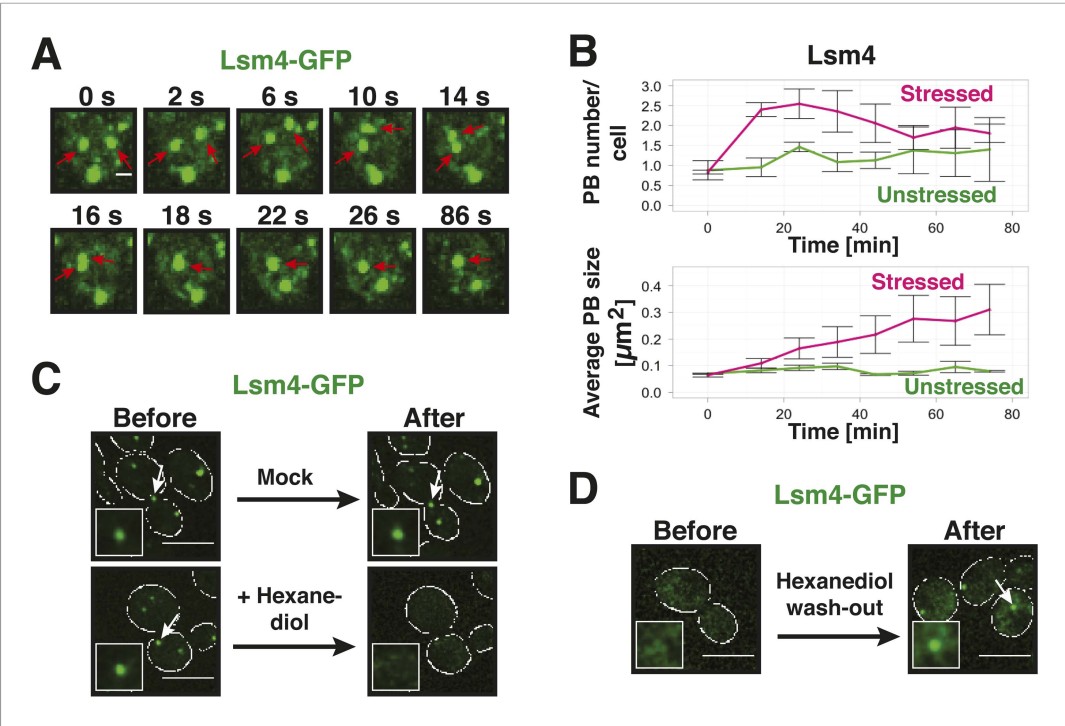

**Figure 3**. Yeast P-bodies are liquid droplets and not aggregates. (**A**) P bodies behave as liquid-like droplets. Fluorescence time-lapse microscopy of stressed cells expressing Lsm4-GFP from the endogenous locus. Time points are indicated in seconds above. Two fusing PBs are indicated by red arrows. White scale bar: 2 μm. (**B**) Quantification of the PB fusion behavior over time using GFP-tagged Lsm4 as a marker. At time point 0, the medium was changed to synthetic medium lacking glucose (red curves). Control cells received complete synthetic medium (green curves). Values given are PB number/cell and the average PB size [μm$^2$]. At last 150 cells were analyzed. Error bars represent SEM. (**C**) P bodies are sensitive to 1,6-hexanediol, suggesting that they are liquid-like. Cells expressing Lsm4-GFP from the endogenous promoter were stressed in medium without glucose for 30 min (Before). 5% 1,6-hexanediol or medium (Mock) was added and the cells were analyzed after 30 min (After). Scale bars: 5 μm. Also see corresponding *Video 4* and related *Figure 3—figure supplement 1*. (**D**) The hexanediol effect is reversible. Stressed cells expressing Lsm4-GFP were treated with 5% 1,6-hexanediol for 1 hr (Before). Hexanediol was washed out with fresh medium and an image was acquired after 12 min (After). Also see corresponding *Video 5*.

The following figure supplement is available for figure 3:

**Figure supplement 1**. The aliphatic alcohol 1,6-hexanediol causes rapid disassembly of P bodies.

aggregates, which result from stress-induced protein misfolding (*Cherkasov et al., 2013*). Thus, we reasoned that stress granules might be functional aggregates that behave in a similar manner as aggregates formed by misfolded proteins. To investigate this, we compared the aggregation behavior of stress granule components and misfolded proteins by time-lapse microscopy. For this purpose, we used yeast strains co-expressing GFP-tagged stress granule proteins (Pbp1 or Nrp1) and mCherry-tagged misfolding-prone proteins (a mutant variant of luciferase or a thermo-labile variant of Ubc9, Ubc9ts) (*Kaganovich et al., 2008*; *Gupta et al., 2011*). Indeed, in cells that were exposed to a robust heat shock of 46°C, stress granule proteins co-localized with luciferase or Ubc9ts in punctate structures (*Figure 5A*, *Figure 5—figure supplement 1*). Co-localization was also observed between Nrp1 and several chaperones, such as Hsp42, Ssa1, and Hsp104 (*Figure 5—figure supplement 2*). This indicates that stress granule proteins and misfolded proteins are co-deposited, as previously suggested (*Cherkasov et al., 2013*).

Despite the fact that protein aggregates and stress granule components showed strong co-localization, stress granules dissolved faster than misfolded proteins during recovery from stress (*Figure 5—figure supplement 3*; *Videos 7–9*). Importantly, cells that had dissolved their stress granules re-entered into the cell cycle (*Videos 7–9*), although they still contained aggregated

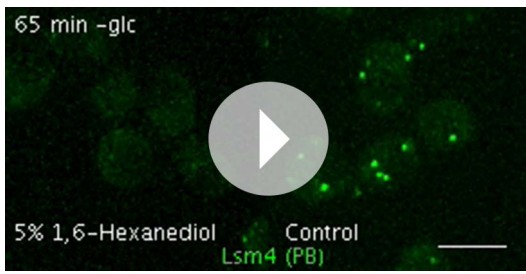

**Video 4.** Hexanediol treatment disrupts P bodies in yeast cells. Cells expressing Lsm4-GFP from the endogenous promoter were stressed in medium without glucose for 30 min. 5% hexanediol or medium (Control) was added and the cells were analyzed by time-lapse fluorescence microscopy. Related to *Figure 3*.

misfolded proteins. This suggests that the presence of misfolded proteins does not prevent re-entry into the cell cycle and that resumption of growth may be coupled to stress granule dissolution. It also implies that stress granule components are less aggregation-prone than the misfolding-prone model proteins used in our study. To investigate this possibility, we repeated the heat shock experiment at 42°C. Indeed, under mild heat-shock conditions only luciferase and Ubc9ts formed visible aggregates, whereas the stress granule component Nrp1 remained diffusely localized (*Figure 5B* and *Figure 5—figure supplement 4*). These findings suggest that stress granule components behave like misfolding-prone proteins, which reversibly aggregate into stress granules, when cells are exposed to robust environmental stress.

Stress granules form under a variety of conditions, and they show different compositions depending on the nature of the inducing stress (*Hoyle et al., 2007*; *Grousl et al., 2009*; *Buchan et al., 2011*). To determine whether stress granules induced by other stresses behave in a similar way, we exposed yeast to glucose depletion. Glucose removal also caused stress granule formation, and this was accompanied by a low level of luciferase aggregation (see *Video 10*). However, in contrast to heat stress conditions, luciferase and stress granule components coalesced into largely distinct structures (*Figure 5C* and *Figure 5—figure supplement 5*). Thus, we conclude that stress granule proteins aggregate under a variety of conditions but that co-aggregation with misfolded proteins is most pronounced during robust heat shock.

A previous report proposed a role for disaggregating chaperones in stress granule dissolution (*Cherkasov et al., 2013*). Therefore, we next tested how protein disaggregases affect stress granule formation. Three different proteins promote the disaggregation of protein aggregates in yeast: Hsp104 and two members of the Hsp110 family, called Sse1 and Sse2 (*Glover and Lindquist, 1998*; *Shorter, 2011*; *Duennwald et al., 2012*; *Rampelt et al., 2012*; *Doyle et al., 2013*). We compared wild-type cells and cells in which these proteins had been inactivated genetically. Deletion of either of the disaggregases led to more pronounced stress granule assembly and a delay in stress granule disassembly (*Figure 5D,E*, *Video 11*). However, inactivation of Hsp104 had the strongest effect. Similar findings were obtained for stress granules induced by glucose depletion (*Figure 5F*). Thus, we conclude that stress granules are functional aggregates and that the components contained in stress granules need to be reactivated by disaggregation before cells can re-enter into the cell cycle. Consistent with this, a recent report showed that cell cycle-associated RNP granules in the multinuclear fungus *Ashbya gossypii* are functional aggregates that are remodeled by chaperones (*Lee et al., 2015*).

## Misfolded proteins can nucleate yeast stress granules
Our findings indicate that during severe stress conditions both misfolded proteins and stress granule components co-aggregate.

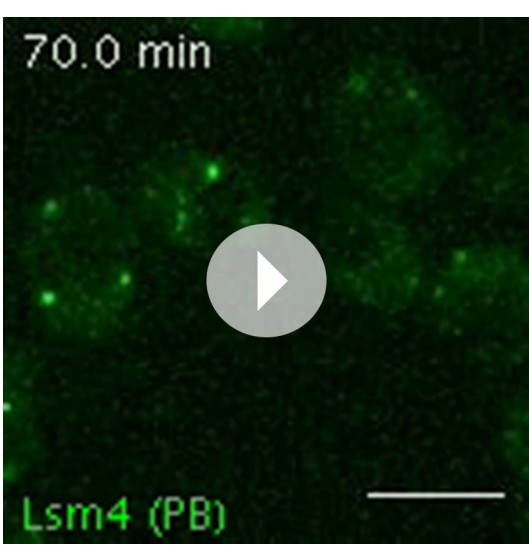

**Video 5.** Reformation of P bodies after hexanediol removal. Glucose starved cells expressing Lsm4-GFP were treated with 5% hexanediol for 1 hr. Hexanediol was washed out with fresh medium and the cells were observed by time-lapse fluorescence microscopy. Related to *Figure 3*.

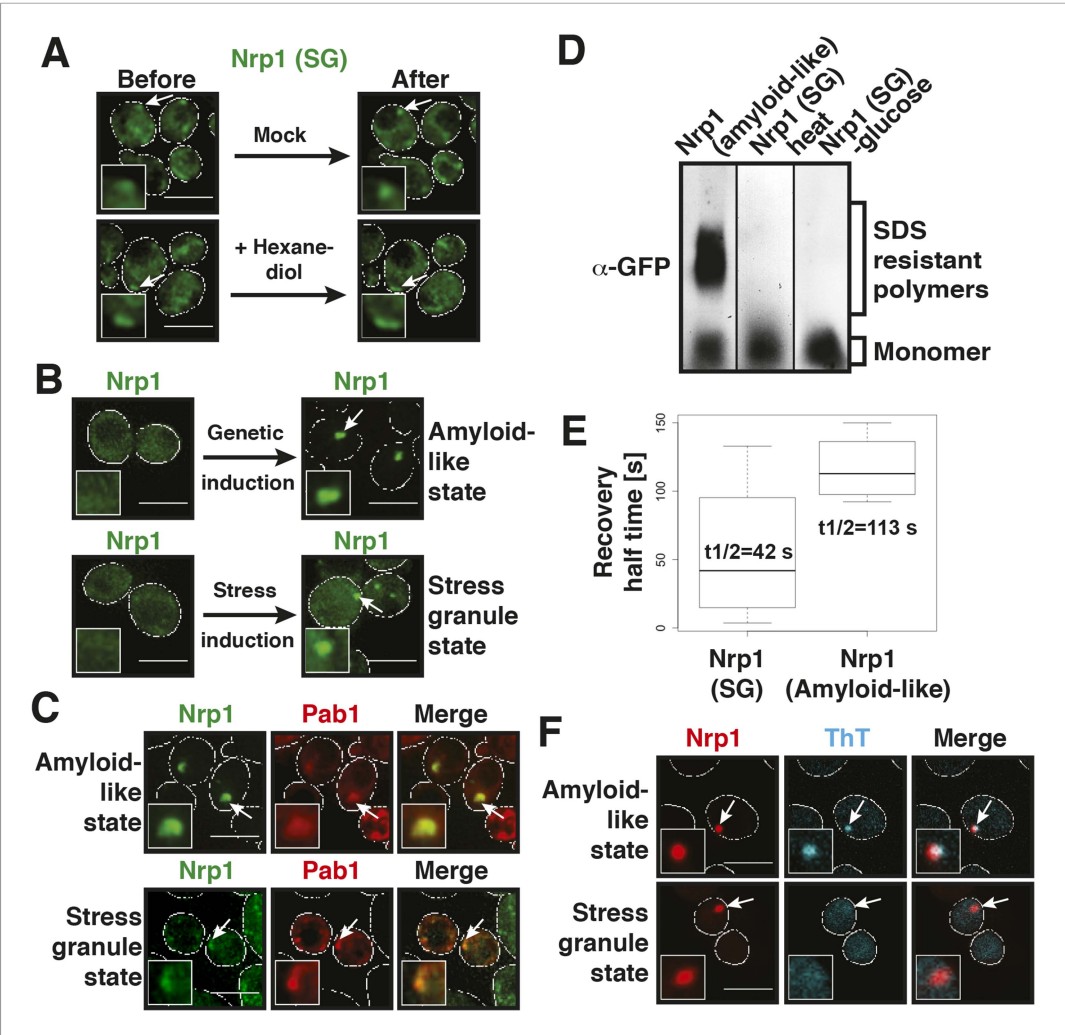

Figure 4. Stress granules have properties of solid aggregates, but do not depend on amyloid-like conversions for their formation. (A) 1,6-hexanediol does not disrupt stress granules. Cells expressing Nrp1-GFP from the endogenous promoter as a marker for stress granules were stressed for 15 min in glucose-deficient medium (Before). An image was taken 6 min after the addition of 10% hexanediol (After). The cells were treated with 10 µg/ml digitonin to make them more permeable to hexanediol. The control (Mock) received only medium +10 µg/ml digitonin. Note that P bodies dissolve within less than 2 min under the same conditions (see *Figure 3—figure supplement 1* and *Figure 4—figure supplement 1*). Also see corresponding *Video 6*. (B) The amyloid-like and stress granule states of Nrp1 are morphologically similar. Fluorescence microscopy of cells expressing GFP-tagged Nrp1 from the endogenous promoter. Nrp1 assembly was induced genetically (amyloid-like state) or through 1 hr glucose starvation (stress granule state). White lines indicate the cell boundaries. Scale bars: 5 µm. (C) Genetically induced and stress-induced Nrp1 assemblies are morphologically similar and recruit Pab1. Fluorescence microscopy analysis of Nrp1-GFP cells expressing mCherry-tagged Pab1. Stress was induced through heat shock at 46°C 10 min. (D) Stress does not induce an amyloid-like state in Nrp1. SDD-AGE of lysates from cells expressing Nrp1-Cerulean from the endogenous promoter. Nrp1 assembly was induced genetically (amyloid-like state) or through stress (stress granule state, SG). Also see related *Figure 4—figure supplement 4*. (E) Analysis of Nrp1 turnover in the stress granule state (induced by glucose depletion) and amyloid-like state. Nrp1-Dendra2 was expressed from a plasmid. The recovery half times of Nrp1 in the SG state (median = 42 s, n = 4) and in the amyloid-like state (median = 113 s, n = 4) are shown. p value = 0.0927 (unpaired Student's *t*-test). Also see related *Figure 4—figure supplement 3*. (F) Stress granules are not stainable with ThT. Cells expressing mCherry-tagged Nrp1 were treated with ThT. Assembly of Nrp1 was induced genetically (amyloid-like state) or through glucose depletion (stress granule state). Also see related *Figure 4—figure supplement 2*.

*Figure 4. continued on next page*

*Figure 4. Continued*

The following figure supplements are available for figure 4:

**Figure supplement 1**. P bodies but not stress granules are sensitive to hexanediol.

**Figure supplement 2**. Stress granule-associated and amyloid-like Nrp1 form different complexes in yeast cell lysate.

**Figure supplement 3**. Analysis of the turnover of Nrp1 in the stress granule and the amyloid-like state.

**Figure supplement 4**. Prion-like stress granule proteins do not transition into an amyloid-like state.

However, despite being deposited in spatial proximity, the two types of proteins do not seem to form mixed aggregates, in particular when exposed to mild stress condition such as glucose depletion.

A previous study suggested that misfolded proteins seed the formation of stress granules (*Cherkasov et al., 2013*). However, it remained unclear whether this is a general mode of stress granule formation or only applies to conditions of robust heat stress. Investigation of a seeding function requires experimental control over the aggregated state of a protein in the context of a living cell. To achieve this, we developed a method based on a self-assembling protein fragment, derived from a viral capsid protein (μNS). This fragment assembles into large spherical particles in cells, and these particles could be visualized by adding GFP to the N terminus (*Figure 6A*). To exclude that these particles are aggregates of misfolded protein, we first tested whether they co-localize with chaperones (Ssa1 and Hsp104). However, chaperones did not associate with μNS particles (*Figure 6—figure supplement 1*), suggesting that the particles are invisible to the protein quality control system.

An earlier study used μNS particles to test for protein–protein interactions in budding yeast (*Schmitz et al., 2009*). In this study, bait proteins were genetically fused to the μNS fragment, which resulted in the presentation of the bait protein on the particle surface. Using this approach, we generated particles that carried either mutated luciferase or Ubc9ts on the surface. We found that both types of particles were recognized by chaperones (*Figure 6B* and *Figure 6—figure supplement 2*), indicating that the misfolded proteins are accessible on the particle surface. Using these particles, we next tested whether misfolded Ubc9ts or luciferase could recruit stress granule components. Under normal growth conditions (25°C), the two stress granule proteins Pab1 or Pub1 were not enriched on the particle surface (*Figure 6C* and *Figure 6—figure supplement 3*, upper panels). However, during a robust heat shock Pub1 and Pab1 accumulated on μNS particles (*Figure 6C* and *Figure 6—figure supplement 3*, lower panels). Naked particles on the other hand did not recruit Pab1 or Pub1 (*Figure 6—figure supplement 4*), indicating that stress granule assembly was specifically triggered by the misfolded proteins. Furthermore, interaction of Pub1 or Pab1 with luciferase was only observed under robust heat shock and not under glucose depletion conditions (*Figure 6—figure supplement 5*). Thus, we conclude that misfolded proteins can nucleate the formation of stress granules under severe heat stress conditions, but not under mild stress conditions or during normal growth.

How do stress granule proteins interact with misfolded proteins? To address this question, we focused on the stress granule protein Nrp1. Nrp1 has an intrinsically disordered PLD, which can undergo amyloid-like conformational conversions (*Alberti et al., 2009*) and shows specific binding to b-isox (*Kwon et al., 2013*). We speculated that the structural flexibility of the PLD could promote the interaction of Nrp1 with misfolded proteins. To test this hypothesis, we mildly

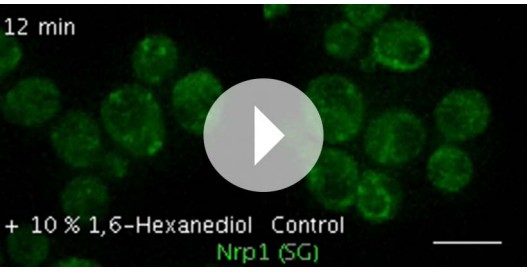

**Video 6.** Hexanediol does not disrupt stress granules. Cells expressing Nrp1-GFP from the endogenous promoter were stressed for 15 min in glucose-deficient medium before the addition of 10% hexanediol. The cells were treated with 10 μg/ml digitonin to make them more permeable to hexanediol. The control received only medium +10 μg/ml digitonin. Note that P bodies dissolve within 2 min under the same conditions (see *Figure 3—figure supplement 1*). Related to *Figure 4*.

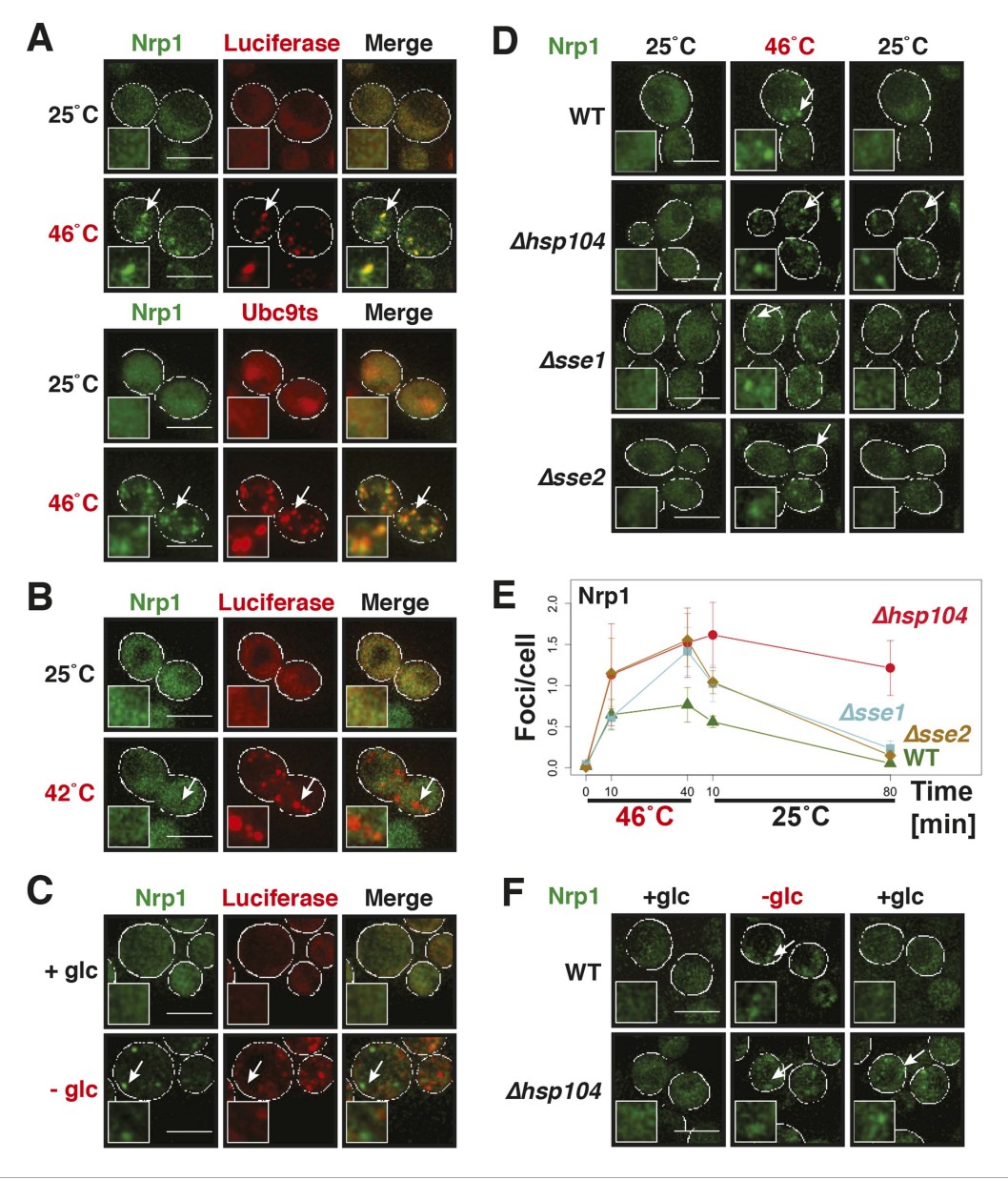

**Figure 5**. Yeast stress granules are functional protein aggregates, which are dissolved by disaggregases. (**A**) The stress granule protein Nrp1 is co-deposited with misfolding-prone proteins during robust heat shock. Fluorescence microscopy of yeast cells expressing Nrp1-GFP from the endogenous locus and mCherry-tagged misfolding-prone proteins (mutated luciferase and Ubc9ts) from a plasmid. The cells were exposed to a 10-min heat shock at 46°C. White lines indicate the cell boundaries. Scale bars: 5 µm. Also see related *Figure 5—figure supplements 1–3* and *Videos 7–9*. (**B**) Nrp1 is less aggregation prone than luciferase. Same as (**A**), except that the cells were exposed to a mild heat shock at 42°C for 10 min. Note that under these conditions only luciferase forms aggregates. Also see related *Figure 5—figure supplement 4*. (**C**) Nrp1 and luciferase coalesce into distinct structures under starvation conditions. Same as (**A**), except that the cells were stressed by glucose starvation for 1 hr (-glc). See corresponding *Video 10*. Also see related *Figure 5—figure supplement 5*. (**D**) Dissolution of stress granules is dependent on disaggregases. Fluorescence time-lapse microscopy of yeast cells expressing GFP-tagged Nrp1 from the endogenous locus. Wild-type cells are compared to strains with genetic deficiencies (Δhsp104, Δsse1, or Δsse2). Shown are images after 40 min at 46°C and during recovery after 80 min at 25°C. Also see *Video 11*. (**E**) Quantification of the number of foci/cell in the strains shown in (**D**). At least 290 cells were analyzed for each strain from three independent experiments. Error bars are SEM. (**F**) Hsp104 is also required for the disassembly of stress granules induced by glucose starvation. *Figure 5. continued on next page*

*Figure 5. Continued*

Wild-type or Δhsp104 cells expressing Nrp1-GFP from the endogenous locus were exposed to glucose starvation for 1 hr (image taken after 40 min at -glc) and observed 80 min after glucose-containing growth medium was added.

The following figure supplements are available for figure 5:

**Figure supplement 1**. The stress granule protein Pbp1 is co-deposited with the misfolding-prone protein Ubc9ts during robust heat shock.

**Figure supplement 2**. The stress granule protein Nrp1 co-localizes with chaperones during robust heat shock.

**Figure supplement 3**. Stress granules dissolve faster than protein aggregates.

**Figure supplement 4**. A mild heat shock does not lead to co-deposition of stress granule components and misfolding-prone proteins.

**Figure supplement 5**. Stress granules and misfolding-prone proteins do not form mixed aggregates under glucose depletion conditions.

overexpressed Nrp1-mCherry in cells carrying luciferase-coated μNS particles. As can be seen in *Figure 6D*, Nrp1 weakly interacted with the particles, even in the absence of stress. Importantly, when we subsequently applied a robust heat shock, a large amount of Nrp1 accumulated on the surface of the particle (*Figure 6D*). This suggests that the interaction between Nrp1 and misfolded proteins is sufficient to ectopically induce stress granule formation. Next, we tested a variant of Nrp1 lacking the PLD. Remarkably, this variant was unable to recognize the luciferase-coated particle (*Figure 6E*). In contrast, the isolated PLD showed robust binding to luciferase particles (*Figure 6E*), demonstrating that the PLD is sufficient for the interaction with misfolded proteins. Thus, PLDs can promiscuously interact with misfolded proteins, and such promiscuous interactions can nucleate the formation of stress granules, in particular under conditions of robust heat stress.

Why do misfolded proteins only nucleate stress granules under robust heat-shock conditions? We reasoned that under mild stress conditions the activity of the cellular chaperone machinery is sufficient to prevent promiscuous interactions between misfolded proteins and stress granule components. Indeed, when Hsp104 was inhibited, a mild heat shock was sufficient to induce the co-aggregation of endogenous Nrp1 and misfolded proteins (*Figure 6F* and *Figure 6—figure supplement 6*).

A mild stress stimulus can protect cells from subsequent severe stress, a phenomenon known as preconditioning (*Parsell and Lindquist, 1993*). Thus, we hypothesized that preconditioning could prevent the co-aggregation of stress granule components and misfolded proteins. We preconditioned yeast cells by increasing the temperatures incrementally from 25°C to 46°C. Under these conditions, Pab1 and Pub1 did not co-aggregate with misfolded proteins but formed assemblies in distinct areas of the cell (*Figure 6G* and *Figure 6—figure supplement 7*). This shows that molecular chaperones constantly work to prevent promiscuous interactions with misfolded proteins and that misfolded proteins only act as scaffolds for stress granules when the capacity of the protein quality control machinery is overrun. It also suggests that preconditioned cells are protected from promiscuous interactions with misfolded proteins, presumably because of the up-regulation of chaperones.

## Stress granule assembly is redundant and adaptable

Our findings suggest that an interaction with misfolded proteins is not required for stress

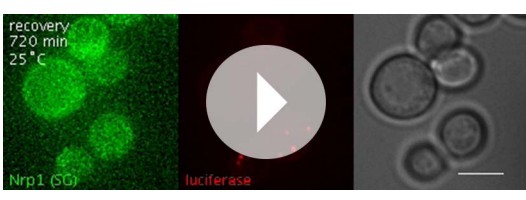

**Video 7.** Stress granule and protein aggregate formation and dissolution in stressed cells. Yeast cells expressing Nrp1-GFP from the endogenous locus were transformed with plasmids for the expression of mCherry-tagged mutant luciferase. Cells were exposed to a robust heat shock at 46°C and then transferred back to 25°C. Note that stress granules dissolve faster than protein aggregates. Related to *Figure 5*.

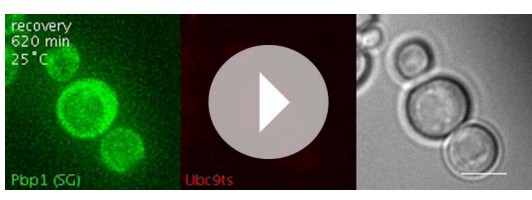

**Video 8.** Stress granule and protein aggregate formation and dissolution in stressed cells. Yeast cells expressing Nrp1-GFP from the endogenous locus were transformed with plasmids for the expression of mCherry-tagged Ubc9ts. Cells were exposed to a robust heat shock at 46°C and then transferred back to 25°C. Note that stress granules dissolve faster than protein aggregates. Related to *Figure 5*.

granule formation. To investigate how stress granules form independently of misfolded proteins, we again focused on Nrp1. We first tested which domains of Nrp1 are required for recruitment to stress granules. As shown in *Figure 6H*, the RBD alone was sufficient for localization to stress granules, whereas the PLD was not. This is consistent with many other studies, which reported that PLDs are dispensable for RNP granule localization (*Reijns et al., 2008*; *Sun et al., 2011*; *Cai and Futcher, 2013*; *Grousl et al., 2013*; *Kruger et al., 2013*; *Bley et al., 2014*). One potential explanation for the observed dispensability of PLDs is that they mediate interactions that are redundant and weak and thus only manifest when these domains are present at high local concentrations. To test

this possibility, we used our particle assay to concentrate stress granule proteins in living cells, thus, creating a molecular environment similar to that in RNP granules. First, we generated particles that carried full-length Nrp1 on the surface. After having confirmed that Nrp1-µNS particles are not recognized by chaperones (*Figure 6—figure supplement 8*), we tested whether Nrp1 could ectopically nucleate the formation of stress granules. Indeed, particle-bound Nrp1 was able to recruit additional Nrp1 molecules from the cytosol, and upon heat shock (*Figure 6I*, bottom panel) and glucose depletion (*Figure 6—figure supplement 9*, bottom panel) stress granules formed on Nrp1 particles. As a next step, we tested which domains of Nrp1 are required for this behavior. We generated two deletion mutants comprising either the RBD or the PLD. We first confirmed that these truncation mutants were not misfolded when presented on the particle surface (*Figure 6—figure supplement 10*). Using these particles, we found that both the PLD and the RBD were able to recruit full-length Nrp1 (*Figure 6J*) and promoted the formation of stress granules (*Figure 6—figure supplement 11* and *Figure 6—figure supplement 12*). Together, these findings indicate that stress granule assembly is highly redundant and that nucleation can proceed in multiple ways, through PLDs or RBDs. Our data further suggest that PLDs work synergistically with RBDs and only function when present at a high local concentration, as during RNP granule assembly.

## Maintenance of yeast P-body integrity requires Hsp104

Our findings so far show that P bodies are liquid-like droplets, in contrast to stress granules, which behave as true aggregates. Because of this distinction, we predicted that P bodies do not co-aggregate with misfolded proteins. Indeed, we found that P bodies showed only marginal co-localization with misfolded proteins in cells exposed to robust heat stress (*Figure 7A* and *Figure 7—figure supplement 1*).

Co-localization with chaperones, such as Hsp42, Ssa1 and Hsp104, was also limited (*Figure 7B* and *Figure 7—figure supplement 2*). We next tested whether P-body formation and dissolution is affected by protein disaggregases. Under normal or mild stress conditions, Hsp104 deficiency did not affect P-body formation. However, upon robust heat stress (46°C), the P-body protein Edc3 assembled into irregular aggregate-like structures (*Figure 7C*), which persisted for extended times (*Video 12*). These aggregate-like structures co-localized with the stress granule marker Pub1 and misfolded proteins (*Figure 7D*), suggesting that Edc3 is mistargeted to stress granules in the absence of Hsp104. Similar findings were made for the P-body protein Lsm4 (*Figure 7—figure supplement 3*). Next, we

**Video 9.** Stress granule and protein aggregate formation and dissolution in stressed cells. Yeast cells expressing Pbp1-GFP from the endogenous locus were transformed with a plasmid for the expression of mCherry-tagged Ubc9ts. Cells were exposed to a robust heat shock at 46°C and then transferred back to 25°C. Note that stress granules dissolve faster than protein aggregates. Related to *Figure 5*.

**Video 10.** The stress granule protein Nrp1 is not co-deposited with misfolding-prone proteins during glucose deprivation. Fluorescence microscopy of yeast cells expressing Nrp1-GFP from the endogenous locus and mCherry-tagged mutants luciferase from a plasmid. Related to *Figure 5*.

generated strains that expressed Hsp104 at different levels. We found that the amount of Edc3 that co-aggregated with misfolded proteins decreased when the expression level of Hsp104 was increased (*Figure 7E*, *Video 13*). This indicates that maintenance of the liquid-like P-body state requires the continuous action of Hsp104 during acute stress and that P-body components are mistargeted to stress granules, when the disaggregation activity is insufficient.

## Mammalian and yeast stress granules have distinct material properties

Do mammalian P bodies and stress granules behave in a similar way as those of yeast? To investigate the properties of P bodies and stress granules in mammalian cells, we generated stable HeLa cell lines expressing GFP-tagged G3PB2 or DCP1a as markers for stress granules or P bodies, respectively, using BAC TransgeneOmics (*Poser et al., 2008*).

We tested mammalian stress granules and P bodies for three characteristics that define a liquid-like compartment (*Hyman et al., 2014*): first, a liquid compartment should be roughly spherical due to surface tension. Second, the components within the compartment should undergo rapid internal rearrangement and third, two liquid droplets should fuse and relax into one droplet. Indeed, the stress granules and P bodies in our cell lines had a characteristic circular shape (*Figure 8A*), as expected for a liquid-droplet state. We also noticed that stress granules, in particular in the early stage of stress exposure, merged and formed larger structures over time (*Figure 8B*, *Video 14*), as did P bodies (*Figure 8B*, *Video 15*). In both cases, the structures rapidly relaxed into more spherical structures, in agreement with a liquid-like state. Next, we applied a technique known as 'half-bleach' to test for internal mobility within the compartment. In this method, roughly half a structure is bleached, and the distribution of the fluorescence within the photo-manipulated structure is then determined over time (*Brangwynne et al., 2009*). The analysis of such a half-bleach event showed that G3BP2 was redistributed rapidly within stress granules from the unbleached to the bleached area (because of the small size we cannot perform a similar experiment for P bodies) (*Figure 8C*). To further investigate the material properties of these RNP granules, we treated HeLa cells harboring P bodies and stress granules with hexanediol. Hexanediol triggered the disintegration of both types of compartments, whereas a control amyloid structure, Q103-GFP, was unaffected (*Figure 8D*). Thus, we conclude that P bodies and stress granules have liquid-like properties in mammalian cells: they turn over rapidly, are spherical, and when they fuse they relax into one spherical assembly. This is consistent with previous fluorescence recovery after photobleaching (FRAP) studies showing that components within mammalian RNP granules turn over rapidly (*Kedersha et al., 2000*; *Andrei et al., 2005*; *Kedersha et al., 2005*; *Aizer et al., 2008*; *Bley et al., 2014*).

Our findings demonstrate that mammalian stress granules are more liquid-like than their yeast counterparts. We therefore hypothesized that in mammalian cells, misfolded proteins do not nucleate stress granules. To investigate this, we analyzed the subcellular distribution of stress granules and misfolding-prone luciferase in stressed HeLa cells. As can be seen in *Figure 8E*, stress granules did not overlap with luciferase aggregates, under conditions of arsenate stress, proteasome inhibition, or heat shock. Thus, we conclude that stress granules are formed in fundamentally different ways in yeast and mammalian cells.

**Video 11.** Dissolution of stress granules is dependent on disaggregases. Fluorescence time-lapse microscopy of yeast cells expressing GFP-tagged Nrp1 from the endogenous locus. Wild-type cells are compared to strains with genetic deficiencies (Δhsp104, Δsse1, or Δsse2). Cells were exposed to a robust heat shock at 46°C and then transferred back to 25°C. Related to *Figure 5*.

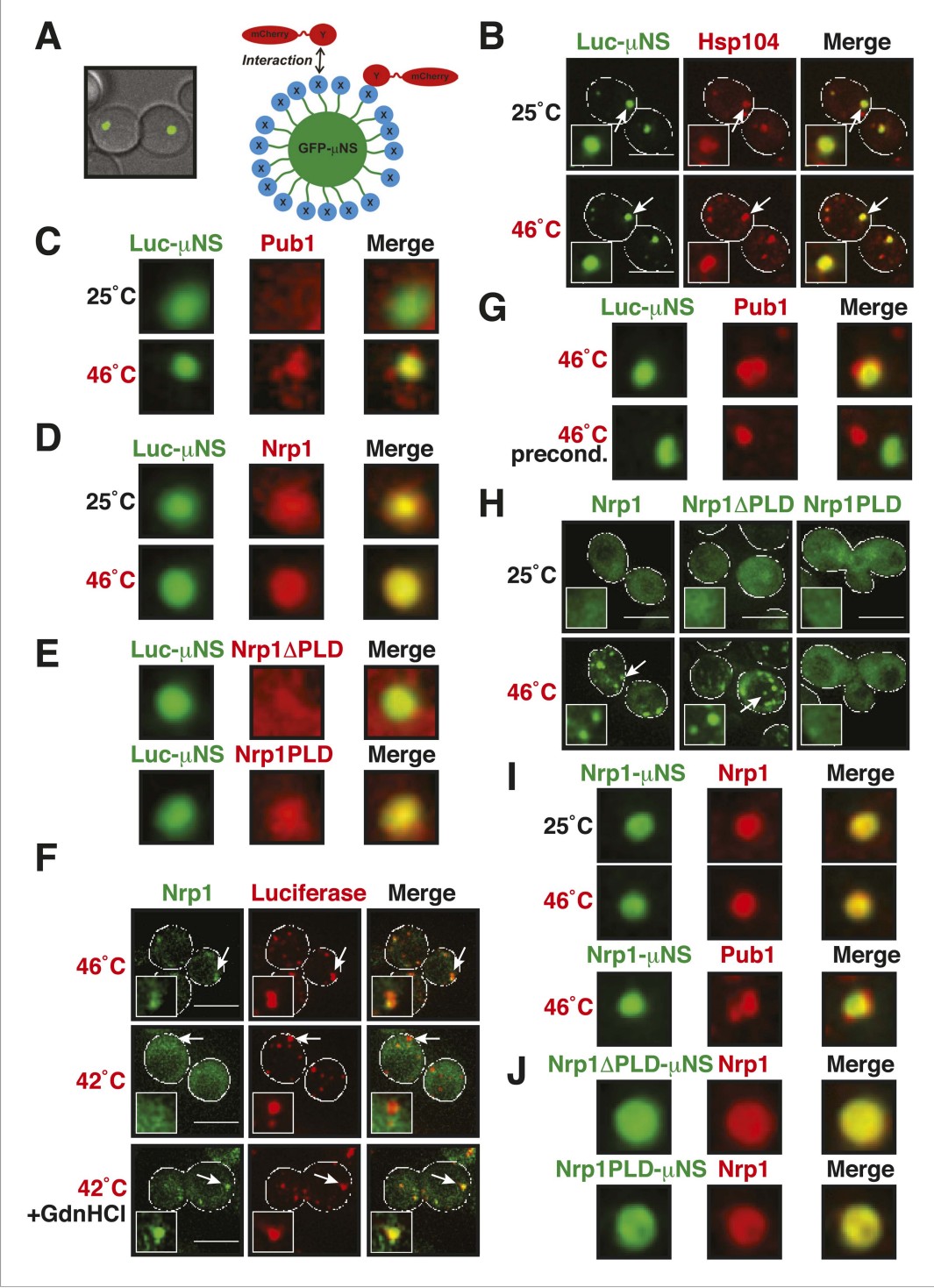

**Figure 6**. Stress granule assembly is redundant and highly adaptable. (**A**) Using a genetically encoded particle to study dynamic interactions in living yeast cells. Left: a fragment of the viral capsid protein μNS comprising the 250 C-terminal amino acids forms self-assembling particles in yeast cells. The particles were visualized using an N-terminal sfGFP tag. Right: interaction assay. Fusion of a protein X to sfGFP-tagged μNS particles allows interaction studies with a mCherry-tagged protein Y. (**B**) μNS particles carrying mutant luciferase on the surface interact with endogenous mCherry-tagged Hsp104. Cells were observed before and after a 10-min heat shock at 46°C. White lines indicate the cell boundaries. Scale bars: 5 μm. Also see related *Figure 6—figure supplements 1, 2*. (**C**) Same as (**B**), except that cells were used in which mCherry-tagged Pub1 was expressed from the endogenous locus. Only one
*Figure 6. continued on next page*

*Figure 6. Continued*

representative µNS particle is shown at high magnification. Note that Pub1 only interacts with luciferase in cells exposed to robust heat stress. Also see related *Figure 6—figure supplements 3–5*. (**D**) Same as (**C**), except that mCherry-tagged Nrp1 was mildly overexpressed from a plasmid carrying an ADH1 promoter. Note that Nrp1 interacts with luciferase already in unstressed cells, and that the amount of Nrp1 accumulating on the particle is strongly increased upon heat stress. (**E**) Same as (**D**), except that the prion-like domain (PLD) of Nrp1 (Nrp1PLD) or a deletion mutant lacking the PLD (Nrp1ΔPLD) was observed at 25°C. (**F**) The cellular chaperone machinery prevents interactions between misfolded proteins and stress granule components. Cells expressing Nrp1-GFP from the endogenous locus and mCherry-tagged mutated luciferase from a plasmid were exposed to a 10-min heat shock at 42°C or 46°C. The cells in the bottom panel were exposed to 3 mM guanidinium hydrochloride (GdnHCl) to inhibit Hsp104. Also see related *Figure 6—figure supplement 6*. (**G**) Same as (**C**), except that the temperature was increased slowly from 25°C to 46°C (preconditioning). Note that preconditioning prevents co-assembly of stress granules and misfolded proteins. Also see related *Figure 6—figure supplement 7*. (**H**) PLDs mediate interactions only when present in high local concentrations. Yeast cells were transformed with plasmids for the expression of GFP-tagged wild-type Nrp1 or deletion mutants lacking the RNA-binding domain (RBD) (Nrp1PLD) or PLD domain (Nrp1ΔPLD). The resulting cells were exposed to heat shock. (**I**) Upon heat shock, stress granules form on µNS particles that present Nrp1 on the surface. Same conditions as (**C**) and (**D**). Also see related *Figure 6—figure supplements 8–9*. (**J**) Same as (**I**), except that mutants lacking the RBD (Nrp1PLD) or PLD domain (Nrp1ΔPLD) were presented on the particle. Note that both mutants are able to recruit full-length Nrp1 at 25°C. Also see related *Figure 6—figure supplement 10–12*.

The following figure supplements are available for figure 6:

**Figure supplement 1**. µNS particles do not interact with Hsp104 or Ssa1.

**Figure supplement 2**. sfGFP-µNS particles carrying mutant luciferase or Ubc9ts on the surface interact with Hsp104 and Ssa1.

**Figure supplement 3**. Misfolded proteins can nucleate stress granule formation under robust heat shock conditions.

**Figure supplement 4**. Control experiment showing that µNS particles do not interact with Pub1 or Pab1.

**Figure supplement 5**. Misfolded proteins do not nucleate stress granule formation in glucose-deprived cells.

**Figure supplement 6**. Chemical inhibition of Hsp104 leads to co-aggregation of misfolded proteins and stress granule components even under mild heat shock conditions.

**Figure supplement 7**. Misfolded proteins and stress granule components do not co-aggregate in preconditioned cells.

**Figure supplement 8**. Control experiment showing that Nrp1-µNS particles do not interact with Hsp104.

**Figure supplement 9**. Nrp1 can nucleate stress granules upon glucose starvation stress.

**Figure supplement 10**. Same as *Figure 6—figure supplement 8* except that mutants lacking the RBD (Nrp1PLD) or PLD domain (Nrp1ΔPLD) were presented on the particle.

**Figure supplement 11**. The PLD as well as the RBD of Nrp1 can nucleate stress granule formation.

**Figure supplement 12**. The PLD or RBD of Nrp1 can nucleate stress granule formation through heterotypic interactions.

## Discussion

In this study, we investigate two types of stress-inducible RNP granules—P bodies and stress granules. We find that these RNP granules have distinct material properties in yeast cells: whereas P bodies show more liquid-like behavior, stress granules exhibit characteristic properties of solid aggregates.

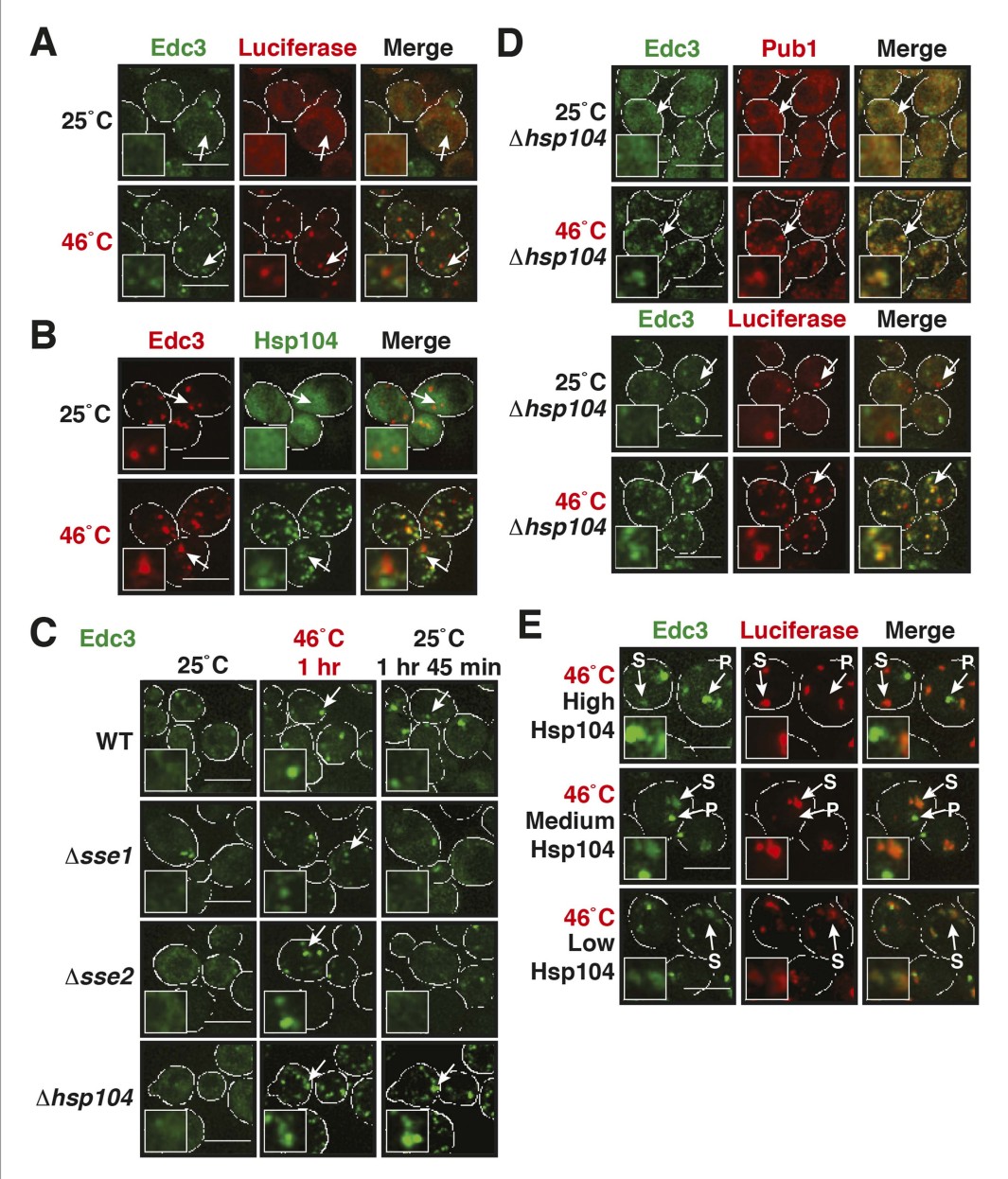

**Figure 7**. Maintenance of yeast P body integrity requires Hsp104. (**A**) The P-body protein Edc3 shows only minor co-localization with misfolded proteins during robust heat stress. Fluorescence microscopy of endogenous GFP-tagged Edc3 and plasmid-expressed mCherry-tagged mutant luciferase. The cells were stressed at 46°C for 10 min. Cell boundaries are indicated in white. Scale bars: 5 μm. Also see related *Figure 7—figure supplement 1*. (**B**) Edc3 shows only limited spatial overlap with Hsp104. Fluorescence microscopy of plasmid expressed mCherry-tagged Edc3 and Hsp104-GFP expressed from the endogenous locus. The cells were subjected to heat stress. Also see related *Figure 7—figure supplement 2*. (**C**) Edc3-positive assemblies show different morphologies and behavior in the absence of disaggregases. Fluorescence time-lapse microscopy of yeast cells expressing GFP-tagged Edc3 from the endogenous locus. Wild-type cells are compared to strains lacking disaggregases (Δhsp104, Δsse1, or Δsse2) after 1 hr heat stress at 46°C and 1 hr 45 min after recovery at 25°C. Also see *Video 12* and related *Figure 7—figure supplement 3*. (**D**) P-body components co-assemble with stress granules in heat-stressed cells in the absence of Hsp104. Yeast cells expressing endogenous Edc3-GFP and endogenous Pub1-mCherry or aggregation-prone mCherry-tagged luciferase from a plasmid in Hsp104-deficient cells were subjected to 1 hr heat stress at 46°C. (**E**) The amount of stress granule-localized Edc3 (denoted with S) decreases with increasing amounts of Hsp104. P denotes P bodies. Yeast cells were used that expressed Edc3 under the endogenous promotor and mCherry-tagged mutated luciferase from a plasmid. Endogenous *hsp104* was deleted and substituted with plasmid-expressed Hsp104

*Figure 7. continued on next page*

*Figure 7. Continued*

under control of a GPD - glyceraldehyde-3-phosphate dehydrogenase (high), ADH1 - alcohol dehydrogenase 1 (medium), or SUP35 (low) promoter. Cells were observed after 1 hr at 46°C. Also see *Video 13*.

The following figure supplements are available for figure 7:

**Figure supplement 1**. P bodies do not co-aggregate with misfolded proteins under robust heat shock conditions.

**Figure supplement 2**. P bodies only show limited co-localization with chaperones under robust heat shock conditions.

**Figure supplement 3**. Lsm4-positive assemblies show different morphologies and behavior in the absence of disaggregases.

---

We further show that the formation of yeast stress granules does not involve amyloid-like conformational conversions. Rather, yeast stress granules resemble amorphous protein aggregates. Assembly of these aggregates depends on interactions with RNAs, and PLDs in RNA-binding proteins contribute to granule formation, by promiscuously interacting with other PLDs or with misfolded proteins. Finally, we show that stress granules have very different properties in mammalian cells, where they have liquid-like characteristics and do not behave as aggregates. In summary, these findings show that RNP granule formation is highly flexible, and that under conditions of acute stress, disaggregating machines play a key role in maintaining the identity and integrity of RNP granules (*Figure 9*).

Cells must respond rapidly to changing environments. This is particularly important for single-celled organisms such as yeast because they are directly exposed to environmental fluctuations. How can a cell respond rapidly and efficiently to stress, while at the same time solving the task of adjusting the activities of numerous proteins and RNAs? An increasing number of studies suggest that this can be achieved by building compartments. Such compartments are condensed phases of proteins and RNAs, which exchange components with the surrounding cytoplasm or nucleoplasm (*Brangwynne et al., 2009*, *2011*; *Hyman and Brangwynne, 2011*; *Hyman and Simons, 2012*; *Li et al., 2012*; *Feric and Brangwynne, 2013*; *Hubstenberger et al., 2013*; *Hyman et al., 2014*). Two types of compartments can be distinguished: compartments for localized biochemistry in which specific chemical reactions occur; and compartments for storage, where macromolecules adopt an inactive, yet re-activatable state. The prediction would be that these two types of compartments have different properties.

Indeed, a large body of work shows that P bodies are active compartments, involved in processing and degrading mRNAs; stress granules on the other hand do not perform biochemical reactions, but store proteins and RNAs (*Anderson and Kedersha, 2009*; *Decker and Parker, 2012*). P bodies are therefore expected to have different properties than stress granules. A P body should allow for the continuous entry and exit of RNAs and proteins, and the components within a P body should be able to rearrange. A liquid phase meets these demands. Stress granules on the other hand do not need to be liquid-like. Their function is to inactivate proteins and RNAs by removing them from the cytoplasm. Thus, a solid-like state is fully compatible with the function of stress granules. Our findings indeed reveal a remarkable distinction between yeast P-bodies and stress granules. Previous studies have also pointed to differences (*Buchan et al., 2008*; *Decker and Parker, 2012*; *Mitchell et al., 2013*; *Shah et al., 2013*). Based on these findings, we propose that yeast P-bodies are liquid-like droplets that form by demixing from the

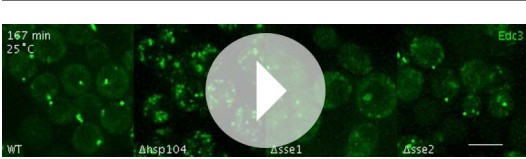

**Video 12.** Edc3-positive assemblies show different morphologies and behavior in the absence of disaggregases. Fluorescence time-lapse microscopy of yeast cells expressing GFP-tagged Edc3 from the endogenous locus. Wild-type cells are compared to strains lacking functional genes for certain disaggregases (Δ*hsp104*, Δ*sse1* or Δ*sse2*). Cells were exposed to a robust heat shock at 46°C and then transferred back to 25°C. Related to *Figure 7*.

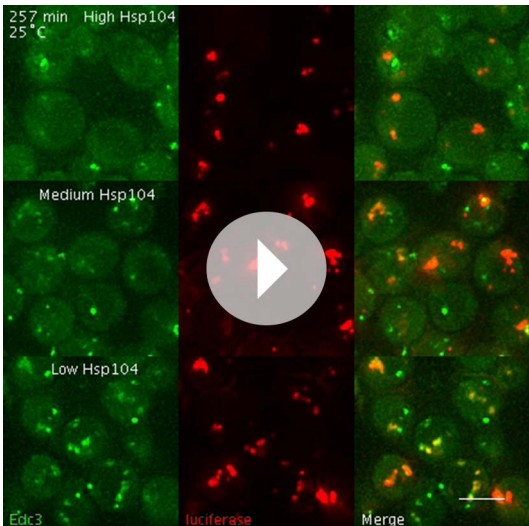

**Video 13.** Edc3 co-aggregates with luciferase in a Hsp104-dependent manner. Endogenous *hsp104* was deleted and substituted with plasmid-expressed Hsp104 under control of a GPD (high), ADH1 (medium), or SUP35 (low) promoter. Cells expressing mCherry-tagged luciferase from a plasmid and GFP-tagged Edc3 from the endogenous locus were exposed to a robust heat shock at 46°C and then transferred back to 25°C. Related to *Figure 7*.

cytoplasm, whereas yeast stress granules form through a liquid–solid phase transition (*Figure 9*). We hypothesize that these findings are specific to yeast and do not apply to mammalian cells. Indeed, studies suggest that mammalian cells have developed other ways of controlling stress granule assembly, primarily through posttranslational modifications of stress granule proteins (*Anderson and Kedersha, 2009*; *Buchan and Parker, 2009*; *Kedersha et al., 2013*; *Wippich et al., 2013*). Future studies will provide further insight into the distinct molecular mechanisms underlying stress granule formation in yeast and mammalian cells.

We used the small organic alcohol 1,6-hexanediol to differentiate between liquid-like and solid-like cellular structures. Hexanediol disperses liquid germ granules in *C. elegans* (*Updike et al., 2011*) and impairs transport across the nuclear pore (*Ribbeck and Gorlich, 2002*; *Patel et al., 2007*), two processes that depend on weak interactions between sticky intrinsically disordered domains. We do not yet understand how hexanediol perturbs RNP granules. However, it should be noted that hexanediol is widely used as an additive in protein crystallization studies. Protein crystallization is typically approached in an empirical manner, and its success often depends on the formation of a liquid

protein phase by phase separation (*Galkin and Vekilov, 2000*; *Chen et al., 2004*; *Dumetz et al., 2008*). It is believed that additives such as hexanediol inhibit or promote the formation of this liquid phase, or otherwise modify its physicochemical properties, and by doing so favor the formation of a crystal. Thus, we speculate that hexanediol perturbs the weak interactions in liquid-like assemblies, leaving stronger interactions that are characteristic of more solid-like structures intact. This suggests that hexanediol could be a useful tool to probe the material properties of cellular structures.

Our observations support the conclusion that yeast stress granules are solid protein aggregates. First, hexanediol disperses P bodies but not stress granules. Second, yeast stress granule components are co-deposited with misfolded proteins. Finally, yeast stress granules are substrates for chaperones and disaggregases. A previous study suggested that stress granule components assemble around misfolded proteins (*Cherkasov et al., 2013*). However, this hypothesis could not be tested directly because it requires control over the nucleation step of protein aggregation. Using an assay to ectopically form protein aggregates, we show that misfolded proteins can indeed nucleate stress granules, in particular when the stress intensity is high and when the overall disaggregation activity is low. However, although yeast stress granules resemble aggregates, they showed no properties of amyloids: they could not be stained with ThT, and we found no evidence that stress granule proteins convert into structures with amyloid-like properties. This suggests that stress granule proteins do not assemble into amyloid-like aggregates under physiological conditions, but rather form amorphous aggregates.

We do not yet know which interactions make yeast stress granules more solid-like. However, we speculate that stress granule proteins carry aggregation-prone domains, which act as stress sensors, as previously also proposed by others (*Cherkasov et al., 2013*). Upon stress, these domains could expose interaction sites that promote assembly into amorphous aggregates. We propose that assembly through these sensor domains is largely specific, as this would provide better control over the assembly process and could facilitate dissolution in the recovery phase. In fact, a similar type of assembly has been observed in starved yeast, where several metabolic enzymes form higher order assemblies (*Narayanaswamy et al., 2009*; *Noree et al., 2010*; *Petrovska et al., 2014*). These enzymes polymerize upon energy

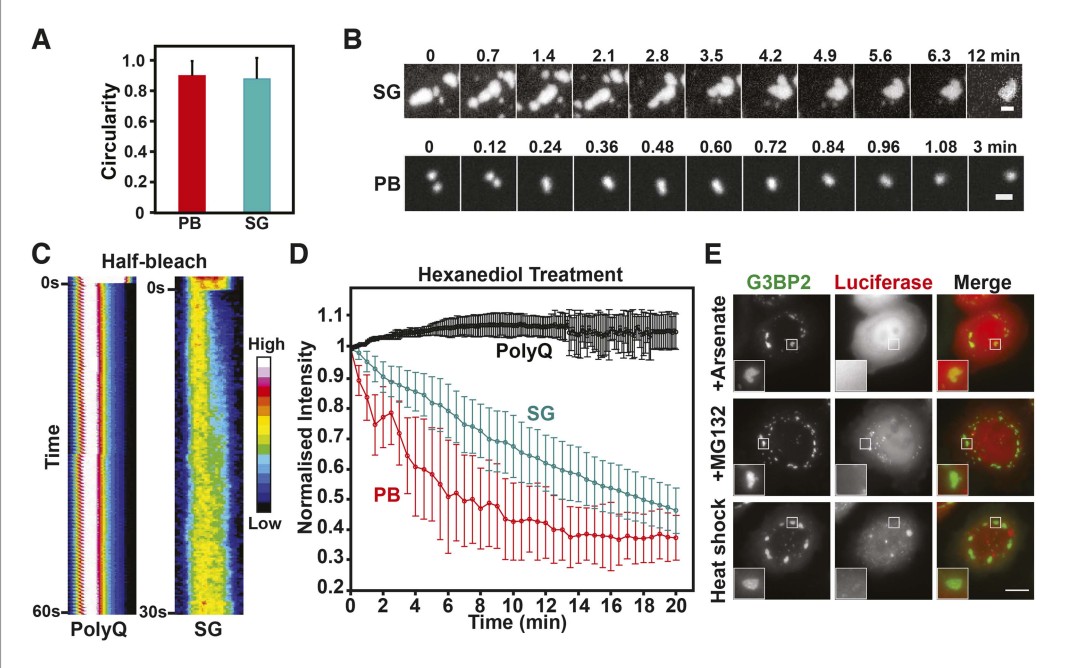

**Figure 8**. Mammalian RNP granules have liquid-like properties and are distinct from yeast stress granules. (**A**) Mammalian P bodies and stress granules are spherical. Quantification of the circularity of stress granules (SG) and P bodies (PB) in HeLa cells, which expressed GFP-tagged DCP1a (PB) or G3BP2 (SG) from BACs (*Poser et al., 2008*). Cells were stressed with 1 mM sodium arsenate for 1 hr. Because of surface tension, liquid-like structures are expected to display a near circular shape (*Hyman et al., 2014*). A perfect circle has a circularity of 1. See 'Materials and methods' for details about how the circularity was determined. The mean and the SD are shown (n=118 PB, n=165 SG). (**B**) P bodies and stress granules show fusion behavior. Time-lapse microscopy of P bodies (DCP1a-GFP) and stress granules (G3BP2-GFP) in arsenate-stressed HeLa cells. Stress granules undergo frequent fusions at the beginning of a stress stimulus and form large spherical structures in cells after extended exposure to stress. Scale bars: 1 μm. Also see corresponding *Videos 14, 15*. (**C**) The stress granule protein G3BP2 shows fast internal rearrangement within stress granules, consistent with a liquid-like state. Kymograph of a stress granule (G3BP2-GFP) induced through arsenate stress and an amyloid (Q103-GFP) aggregate in HeLa cells after a half-bleach event (*Brangwynne et al., 2009*). PolyQ aggregates are used as a control for a solid-like structure. Note that only stress granules but not polyQ assemblies show a redistribution of fluorescence from the bleached to the unbleached area (from left to right). (**D**) P bodies and stress granules but not solid-like polyQ aggregates are sensitive to hexanediol. Quantification of the effect of 3.5% 1,6-hexanediol treatment on stress granules and P bodies induced through arsenate stress, and Q103 aggregates. The normalized mean fluorescence intensities of the structures are plotted over the duration of the treatment. Error bars are SD (n=5 PolyQ, n=31 SG and n=22 PB). (**E**) Mammalian stress granules do not co-aggregate with misfolded proteins. HeLa cells expressing BAC-encoded G3BP2-GFP (SG) were transfected with a plasmid coding for mutant luciferase-mCherry. Formation of stress granules was induced by arsenate stress (2 hr), proteasome inhibition (10 μM MG132, 3 hr) or heat stress (43°C, 2 hr). Scale bar: 10 μm.

depletion, and when yeast cells are replenished with nutrients, they disassemble within minutes. How enzyme assembly can be specific in the crowded environment of a cell is still unclear, but it probably involves interactions via protein surfaces that are sterically and electrostatically compatible. Thus, we propose that yeast stress granules are functional aggregates, which assemble through a controlled process that is driven by a range of specific and promiscuous interactions between proteins and RNAs.

Previous studies proposed an important role for prion-like low-complexity domains in RNP granule assembly. This was based on the observation that these domains can undergo conformational conversions into amyloid-like fibers in a cell-free system (*Han et al., 2012*; *Kato et al., 2012*). Assembly into these structures was accompanied by the formation of hydrogels, and these hydrogels could reversibly interact with other prion-like proteins in a homotypic or heterotypic manner. Accordingly, a model for RNP granule assembly was proposed that put a strong emphasis on the ability of PLDs to assemble into amyloid-like protein scaffolds (*Han et al., 2012*; *Kato et al., 2012*). The problem with this

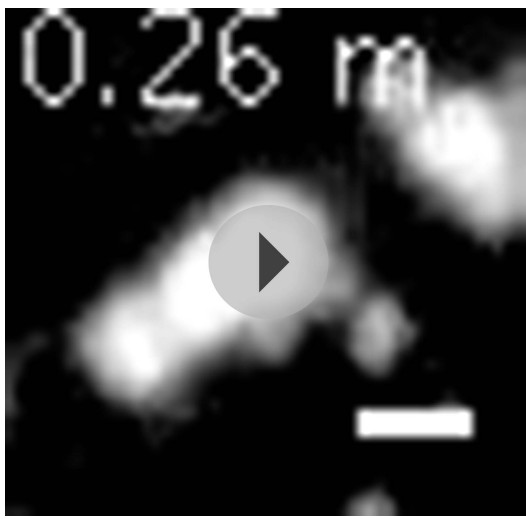

**Video 14.** Stress granules show extensive fusion behavior and form more spherical structures with time. HeLa cells expressing BAC-encoded G3BP2 were observed by time-lapse microscopy. The cells were stressed through arsenate. Related to *Figure 8*.

model, however, is that hydrogels only rearrange over long timescales, and thus can hardly provide the dynamic properties that are characteristic of RNP granules in living cells (*Weber and Brangwynne, 2012*; *Malinovska et al., 2013*; *Bley et al., 2014*). To account for this, it was suggested that these amyloid-like fibers are dynamic in vivo, due to regulation by posttranslational modifications (*Han et al., 2012*; *Kato et al., 2012*; *Kwon et al., 2013*). However, evidence for dynamic fibers in living cells is still lacking.

We used self-assembling particles to locally concentrate prion-like proteins in living cells, creating a molecular environment that resembles that in RNP granules. Using this assay, we find that prion-like proteins promote RNP granule formation, but independently of amyloid-like conversions. Thus, above a certain critical concentration, PLDs can promote a phase transition of RNA-binding proteins into RNP granules. Based on these findings, we favor the following two-step model of RNP granule formation: First, RNA-binding proteins bind to RNAs through multivalent interactions via RBDs, forming complexes of RNA and protein; these RNP complexes

may already reach a relatively large size. In the second step, these RNP complexes condense into RNP granules driven by promiscuous interactions between PLDs and further associations between RNAs and RBDs. In this model, the PLDs primarily assist the final coalescence step, thus, promoting the condensation of RNPs into large compartments (*Figure 9*). In agreement with this, recent findings indicate that PLDs are not essential for RNP granule formation per se but rather promote the fusion of RNP granules and their enlargement into bigger RNP compartments (*Shiina and Nakayama, 2014*). However, the function of PLDs may also depend on their specific local environment: In P

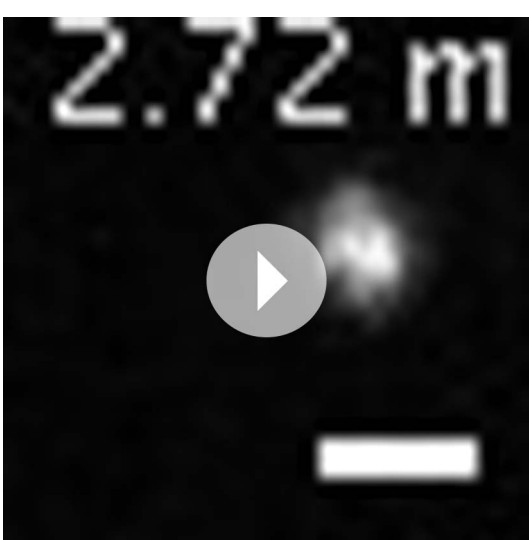

**Video 15.** Two P bodies fuse and rapidly relax into a spherical shape. Time-lapse microscopy of P bodies (labelled by BAC-encoded DCP1a-GFP) in arsenate-stressed HeLa cells. Related to *Figure 8*.

bodies, the collective properties of PLDs may be important for maintaining a dynamic, liquid-like state, whereas in solid assemblies such as stress granules, PLDs may act as molecular glue to connect RNA-binding proteins with other RNA-binding proteins or misfolded proteins. Thus, PLDs may function as adaptable interaction domains, which do not undergo conversions into structurally well-defined amyloid-like states, but mediate promiscuous interactions with many binding partners in their local environment. Such behavior would be highly favorable, because it allows a high degree of flexibility during compartment formation.

We found that in heavily stressed yeast cells, the continuous action of protein disaggregases is required to keep P bodies in a liquid-like state. Interference with protein disaggregation caused the mislocalization of P-body components to stress granules (*Figure 7C–E*), suggesting that P-body proteins and potentially many other proteins can enter into a stress granule state, but are normally prevented from doing so through molecular chaperones such as Hsp104. This implies that the molecular composition of

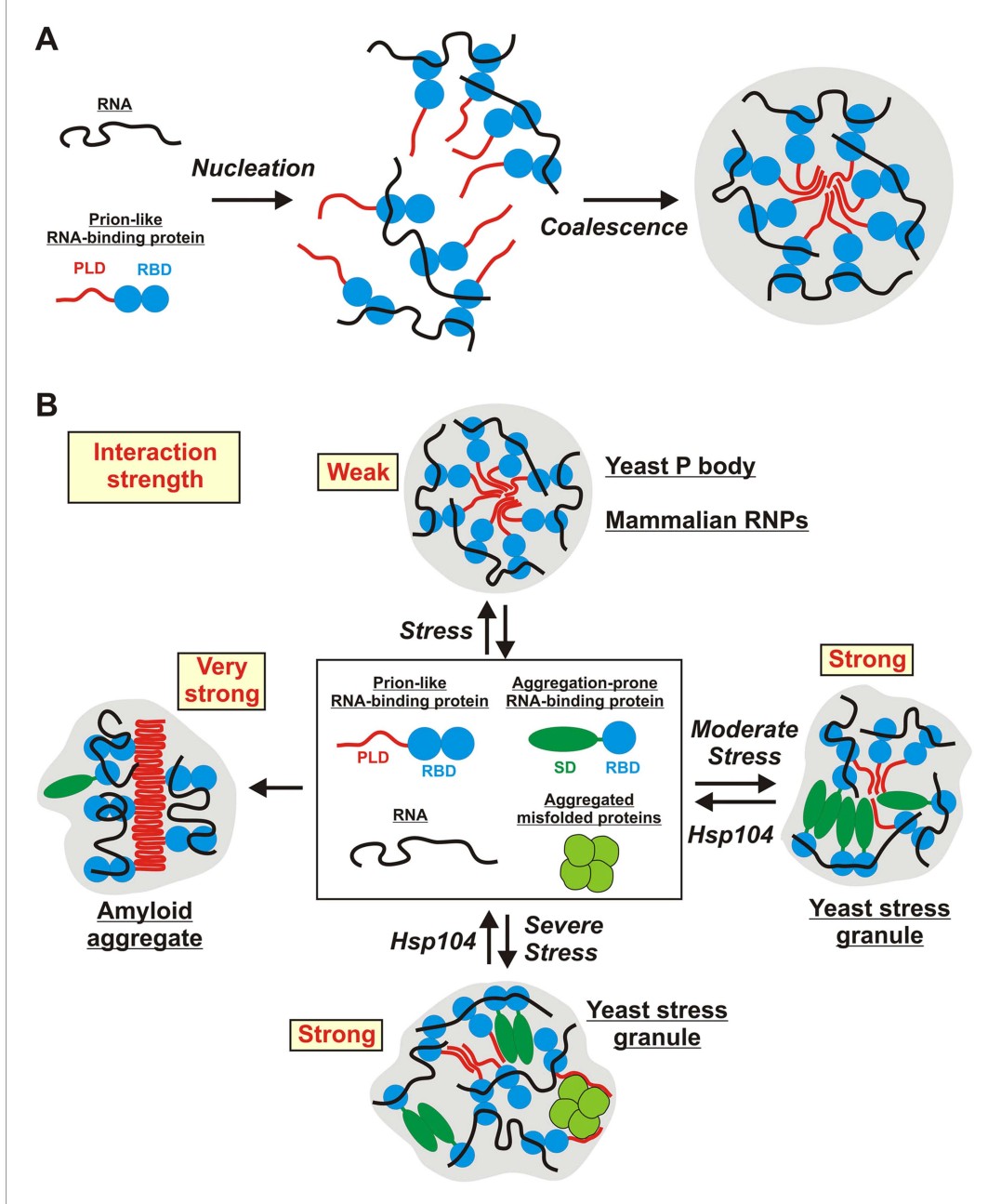

**Figure 9**. Schematic illustrating the potential molecular mechanisms underlying the formation of stress-inducible RNP granules. (**A**) RNA-protein (RNP) granule formation as a two-step assembly process. In the first step, RNAs and RNA-binding proteins associate to form large RNP complexes (nucleation step). In the second step, these RNP complexes coalesce into larger compartments through additional RNA-mediated interactions, but primarily through PLD-mediated weak binding events (coalescence step). PLDs are indicated in red and RBDs indicated in blue. (**B**) Stress-inducible RNP granules have different compositions, which affect their dynamic and material properties. The presumed average interaction strength is indicated in red. Weak interactions increase the vulnerability to hexanediol. An aggregation-prone stress sensor domain (SD) is indicated in dark green.

stress granules is also a function of the cellular disaggregation activity and that yeast cells try to keep a fraction of the cytoplasm in a dynamic liquid-like state, presumably to maintain their viability.

We found that the two Hsp110 disaggregases Sse1 and Sse2 make a significant contribution to the cellular disaggregation activity, but that Hsp104 adopts an essential function, without which yeast cells

cannot dissolve stress granules. In contrast to yeast, mammalian cells do not express a Hsp104 homolog in the cytosol. Therefore, we speculate that the ability of yeast to enter into a solid stress granule state has co-evolved with Hsp104, providing a potential explanation for why yeast and mammalian stress granules have different material properties. In contrast to mammalian organisms, sedentary yeast cells cannot escape from harsh environments. We conjecture that for yeast, formation of solid stress granules is an effective way of preserving the pre-stress state of the cytosol and that a selected set of yeast proteins and RNAs has been modified by evolution to aggregate upon stress. Consistent with this, a recent study showed that the yeast proteome has a higher overall aggregation propensity than the human proteome (*Albu et al., 2014*). When the stress subsides, Hsp104 reactivates the proteins and RNAs preserved in solid stress granules, thus promoting re-entry into the cell cycle. We note that this situation is reminiscent of dormancy, as for example in spores or seeds, where an entire organism enters into a resting state by solidifying its macromolecular components (*Parry et al., 2014*). Future work will show whether controlled entry into a solid phase can provide general protection to a cell in stressful environments.

## Materials and methods

### Cloning procedures

Cloning procedures were performed as described previously using the Gateway system (*Alberti et al., 2007*, *2009*). For a list of plasmids see *Supplementary file 1*.

### Yeast genetic techniques, strains, and media

The media used were standard synthetic media or rich media containing 2% D-glucose. The yeast strain backgrounds were W303 *ADE+* (*leu2-3,112*; *his3-11,-15*; *trp1-1*; *ura3-1*; *can1-100*; [*psi-*]; [*PIN+*]) or BY4741 (*his3Δ1*; *leu2Δ0*; *met15Δ0*; *ura3Δ0*; [*psi-*]; [*PIN+*]). Yeast gene deletions were performed using a PCR-based approach (*Gueldener et al., 2002*). C-terminal tagging of yeast genes was performed as described previously (*Sheff and Thorn, 2004*). The induction of an amyloid state is a concentration-dependent nucleation process. Thus, the prion-like state in Lsm4 was induced by overexpressing untagged Lsm4 in a [*PIN+*] strain that expressed GFP-tagged Lsm4 from the endogenous promoter. The prion-like state in Nrp1 was induced by using a [*PIN+*] strain expressing the Nrp1PLD (*Alberti et al., 2009*). For a list of strains see *Supplementary file 1*.

### Cell culture of HeLa BAC cell lines

HeLa cells were cultured in DMEM (Dulbecco's modified eagle's medium) supplemented with 10% FBS (fetal bovine serum) and penicillin–streptomycin (all Gibco Life Technologies, United Kingdom). Cells were maintained at 37°C in a 5% $CO_2$ incubator. HeLa Kyoto cells containing G3BP2-GFP or DCP1a-GFP BAC constructs were used to visualize stress granules or P bodies, respectively (*Poser et al., 2008*). Transient transfection of HeLa cells with pcDNA3.1-Q103 was performed using Jet Prime transfection reagent (Polyplus, France). Transfection with pcDNA3.1-Luciferase was performed using Lipofectamine 2000 (Invitrogen, Carlsbad, California).

### Wide-field fluorescence microscopy of yeast cells

Yeast cells were grown in cultures of 50–100 ml at 25°C to an $OD_{600}$ not higher than 0.5. The yeast cells were then immobilized on concanavalin A (Sigma Aldrich, St.Louis, Missouri) -coated precise glass bottom dishes (MatTek, Ashland, Massachusetts). Microscopy was performed using a DeltaVision microscope system with softWorx 4.1.2 software (Applied Precision, United Kingdom). The system was based on an Olympus IX71 microscope (Olympus, Japan), which was used with a UPlanSApo 100 × 1.4 numerical (NA) oil objective. The images were collected with a Cool SnapHQ camera (Photometrics, Tucson, Arizona) and a pixel size of 0.13 µm. Heat shock experiments were performed using a Warner heating chamber (Warner instruments, Hamden, Connecticut). When indicated, 5–10% of 1,6-hexanediol (Merck, Germany) solution or 100 µg/ml cycloheximide (AppliChem, Germany) was added to the medium to perturb RNP granule integrity. To inhibit Hsp104, guanidinium hydrochloride (GdnHCl) was added to the medium to a final concentration of 3 mM, 3 hr before imaging. All images were deconvolved using standard softWorx deconvolution algorithms (enhanced ratio, high-noise filtering). Shown images are maximum intensity projections of 8–14 individual images. Figures show

representative cells. The cell boundaries were introduced by thresholding the bright-field image and overlaying it with the fluorescence image. In all comparative experiments, we used four chamber dishes (MatTek), allowing us to image four different conditions in the same experiment.

Analysis of the foci-to-cytoplasm ratio of yeast cells (*Figure 1B*) was performed by manually defining regions of interest and measuring their fluorescence intensity using Fiji (*Schindelin et al., 2012*). The P-body number and size analysis (*Figure 3B*) was performed using Fiji image analysis tools. The image was segmented by thresholding and then analyzed using the analyze particle tool (size: 0–infinity, circularity: 0–1). The number of total cells was determined by manually counting cells in the bright-field channel using the Fiji cell counter plugin.

## Confocal microscopy of ThT-stained yeast cells

Cells were grown in 50 ml cultures while shaking at 180 rpm to an $OD_{600}$ not higher than 0.5. Cultures were then harvested and resuspended in a 10 mM Tris/EDTA (ethylenediaminetetraacetic acid) buffer (pH 7) containing 30 μM ThT (Sigma Aldrich) and incubated for 20 min. Subsequently, the cells were washed three times in Tris/EDTA buffer without ThT and then resuspended in media. Cells were then immobilized on 1% agar pads. Microscopy was performed on a single photon point scanning confocal Olympus Fluoview 1000 microscope (Olympus). ThT was excited with a 405-nm diode laser. Emission of ThT was recorded in the 480–540 nm range. For co-localization studies, we used cells that expressed mCherry-tagged proteins. mCherry fluorescence was excited with a 561-nm DPSS laser, and emission was recorded in the 570–670 nm range. Cells were scanned in one z-plane using a UPlanSApo 60 × 1.35 numerical (NA) oil objective and a scan speed of 8 μs/pixel. Images were acquired with a pixel resolution of 0.103 μm.

## FRAP of yeast cells

Yeast cells were grown in cultures of 50–100 ml at 25°C to an $OD_{600}$ not higher than 0.5. The cells were immobilized in precise glass bottom dishes using concanavalin A coating. Images were acquired using an Olympus IX81 inverted stand microscope (Olympus) with a spinning disk scan head Yokogawa CSU-X1 (5000 rpm). The system was used with an Olympus UPlanSApo 100 × 1.4 oil objective. Acquired images had pixel sizes of 0.081 μm. The images were collected with an Andor iXon EM+ DU-897 BV back-illuminated EMCCD (Andor, Ireland). For imaging of Dendra2, we used a triple-band dichromatic mirror: T-405/488/561. Images before and after photoconversion were acquired as z-stacks with 0.7-μm spacing and a time interval of ~1 s. The Dendra2 signal was converted in a 1 × 1 pixel area by using the 405-nm diode laser at 6% intensity in 400 repeats for 100 μs each. For image segmentation, we used the 3D tracking tool of the Imaris software (Bitplane, Switzerland). The signal from three areas was obtained for each time point: the photoconverted/bleached foci ($I_F$), a reference focus in a neighboring cell ($I_R$) and an area of equal size in the background ($I_B$). The fluorescence signal in the foci was normalized as follows (*Carisey et al., 2011*):

$$I_{F-pre} = \frac{\sum_{t=0}^{t_{bleach}-1} I_F(t) - I_B(t)}{f_{prebleach}},$$

$$I_{R-pre} = \frac{\sum_{t=0}^{t_{bleach}-1} I_R(t) - I_B(t)}{f_{prebleach}},$$

$$I_{F\_norm}(t) = \frac{\frac{I_F(t) - I_B(t)}{I_{F-pre}}}{\frac{I_R(t) - I_B(t)}{I_{R-pre}}}.$$

Since yeast cells are small, we had to ensure that photoconversion in one cell did not affect the reference foci in the neighboring cell. Therefore, we estimated the decay kinetics of the reference foci by fitting the following equation to the post-bleach period of $I_R$:

$$y = A*e^{(p*x)},$$

where $p$ is the time constant and $A$ is the fluorescence intensity. The average fluorescence level before the photoconversion was deduced by extrapolating the fitted decay curve, yielding $I_{R\_estimate}$. If $I_{R\_estimate}$ was smaller than $I_{R\_measured}$, it indicated that fluorescence loss in the reference foci occurred through photoconversion. These cells were excluded from the analysis. To reveal the half-time of recovery, we fitted the values of $I_{F\_norm}$ with:

$$y = A(1 - e^{-p*x}),$$

where $p$ is the time constant and $A$ is the fluorescence intensity.

The half time $t_{1/2}$ was calculated using:

$$t_{1/2} = \frac{\log(0.5)}{-p}.$$

Data were analyzed, tested for statistical significance, and plotted using R software. Boxes in boxplots extend from the 25th to 75th percentiles, with a line at the median.

## Fluorescence microscopy of HeLa cells

Live HeLa cells were imaged using the DeltaVision imaging system with softWorx 4.1.2 software (described above). The system was used with a Plan Apo 60 × 1.42 NA oil immersion objective. 15 sections with 200-nm spacing were acquired and the maximum intensity projections were created in Fiji. If indicated, 3.5% of hexanediol solution was added to the culture medium to perturb P body and stress granule integrity. Measurement of the circularity of P bodies and stress granules (*Figure 8A*) was performed using a custom-made Matlab routine (see *Supplementary file 2*) from maximum intensity projections, according to the following equation:

$$Circularity = \frac{4\pi Area}{(Perimeter)^2},$$

where the value of circularity is 1 for a perfect circle and the value decreases as it deviates from the circular shape. Microscopy to determine the circularity of granules was performed using an Andor spinning disk confocal microscope as described above with a UPlanSApo 100 × 1.4 oil objective.

## Half-bleach experiment in HeLa cells

The half-bleach (*Brangwynne et al., 2009*) in *Figure 8C* was performed at a pixel resolution of 0.08 µm using a spinning disk confocal microscope as described above. Photo-bleaching and imaging was performed with the 488-nm laser. Structures were photo-bleached for 1.2 ms within a region of 5 × 5 pixels.

## SDD-AGE

SDD-AGE analysis was used to separate amyloid polymers from monomers in whole cell lysates. Yeast cells were grown in cultures of 100 ml at 25°C to an $OD_{600}$ of 0.5. The cultures were harvested and washed once with water. The pellets were resuspended in 300 µl ice-cold lysis buffer (50 mM Tris, pH 7.5; 150 mM NaCl; 2.5 mM EDTA; 1% (vol/vol) TritonX-100; 0.33 mM PMSF(phenylmethylsulfonyl fluoride); 6.7 mM NEM; 1.25 mM benzamidine; 10 µg/ml pepstatin; 10 µg/ml chymostatin; 10 µg/ml aprotinin; 10 µg/ml leupeptin; 10 µg/ml E-64) and added to ice-cold glass beads (425–600 µm) (Sigma–Aldrich). Cells were lysed using mechanical disruption (Tissue Lyser II, Qiagen, Netherlands) at 25 Hz for 15 min. Unwanted cell debris and beads were removed by centrifugation and the supernatant was used for SDD-AGE analysis. The supernatants were adjusted for equal protein concentrations and mixed 4:1 with 4 × Sample buffer (40 mM Tris acetic acid, 2 mM EDTA, 20% glycerol, 4% SDS, bromophenol blue). Samples were incubated for 10 min at room temperature and loaded onto a 1.5% agarose gel containing 0.1% SDS in 1 × TAE/0.1% SDS running buffer. The gel was run at low voltage (less than 80 V) to prevent the gel

from heating up. Proteins in the gel were detected by immunoblotting with a GFP-specific antibody (Roche, Switzerland). The exposure time was the same for all immunoblots shown in one experiment.

## SEC

SEC was used to separate cellular complexes according to their size. Cells were grown in cultures of 50 ml at 25°C to an $OD_{600}$ not higher than 0.5. The cultures were harvested and washed once with water. The pellets were resuspended in 500 µl ice-cold lysis buffer (50 mM Tris, pH 7.5; 150 mM NaCl; 2.5 mM EDTA; 1% (vol/vol) TritonX-100; 0.4 mM PMSF; 8 mM NEM; 1.25 mM benzamidine; 10 µg/ml pepstatin; 10 µg/ml chymostatin; 10 µg/ml aprotinin; 10 µg/ml leupeptin; 10 µg/ml E-64) and ice-cold glass beads were added (425–600 µm). Cells were lysed using mechanical disruption (Tissue Lyser II) at 25 Hz for 20 min. To pellet cell debris and beads, samples were centrifuged and the supernatant was applied to a SpinX-Centrifuge Filter (0.22 µm cellulose acetate, Sigma-Aldrich). After centrifugation for 5 min at 13.000 rpm at 4°C, protein concentration of the supernatant was measured using the Bradford protein assay (BioRad, Germany). Equal amounts were loaded onto the column. To efficiently resolve the protein complexes, we used the TSK-Gel G6000PWXL (Tosoh Bioscience, Japan) column with a separation range between 40 kDa and 8.000 kDa. The column was equilibrated in running buffer (2 × PBS (phosphate-buffered saline)) and calibrated with different standard proteins: Tyroglobulin (660 kDa), Ferritin (440 kDa), Catalase (240 kDa), and BSA (bovine serum albumin) (67 kDa). Subsequently, a dot blot assay was performed using a vacuum slot blot device (Whatman, United Kingdom). The fractions were collected on protein-binding nitrocellulose membrane (Protran B 85, Whatman) and the resulting membrane was used for immunodetection with an anti-GFP antibody.

## Acknowledgements

We thank several members of the MPI-CBG for critical reading of the manuscript. We thank Daniel Kaganovich for a plasmid containing mutant luciferase and Cammie Lesser for the GFP-µNS plasmid. The light microscopy facility and the chromatography facility of the MPI-CBG are acknowledged for expert technical assistance. We acknowledge funding from the Max Planck Society, the Dresden International Graduate School (DIGS-BB), and the Alexander von Humboldt Foundation.

## Additional information

### Funding

| Funder | Grant reference | Author |
| --- | --- | --- |
| Max-Planck-Gesellschaft | | Sonja Kroschwald, Shovamayee Maharana, Daniel Mateju, Liliana Malinovska, Elisabeth Nüske, Ina Poser, Doris Richter, Simon Alberti |
| Alexander von Humboldt-Stiftung | | Shovamayee Maharana |
| Technische Universität Dresden | Dresden International Graduate School for Biomedicine and Bioengineering | Sonja Kroschwald, Daniel Mateju, Liliana Malinovska, Elisabeth Nüske |

The funders had no role in study design, data collection and interpretation, or the decision to submit the work for publication.

### Author contributions

SK, Conception and design, Acquisition of data, Analysis and interpretation of data, Drafting or revising the article; SM, DM, LM, EN, Conception and design, Acquisition of data, Analysis and interpretation of data; IP, Acquisition of data, Contributed unpublished essential data or reagents; DR, Acquisition of data, Analysis and interpretation of data; SA, Conception and design, Analysis and interpretation of data, Drafting or revising the article

## Additional files

**Supplementary files**

• Supplementary file 1. List of yeast strains used in this study.

• Supplementary file 2. Matlab routine for measuring the circularity of granules.

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
