## [Decision Letter]

Thank you for sending your work entitled “Promiscuous interactions and protein disaggregases determine the material state of stress-inducible RNP granules” for consideration at *eLife*. Your article has been favorably evaluated by Randy Schekman (Senior editor), Jeffery Kelly (Reviewing editor), and two reviewers, one of whom, Aaron Gitler, has agreed to reveal his identity.

The Reviewing editor and the reviewers discussed their comments before we reached this decision, and the Reviewing editor has assembled the following comments to help you prepare a revised submission.

Non membrane-delimited subcellular compartments play critical cellular functions essential for life. Two such subcellular compartments are processing bodies (P-bodies) and stress granules (SGs), which are composed of RNAs and RNA-binding proteins. Many of the protein constituents of SGs and P-bodies contain domains resembling yeast prions (called prion-like domains). Are SGs and P-bodies merely protein aggregates or do they represent structures with other physical and chemical properties? The manuscript by Kroschwald et al. addresses the physicochemical nature of RNP bodies and uses genetic, chemical, and microscopic imaging approaches. The paper focuses on P bodies and stress granules in budding yeast, and mammalian tissue culture cells are also briefly considered. The authors convincingly show that there are different types of bodies that form in yeast: dynamic liquid-like structures that are SDS soluble and which can also be dissolved in vivo by the application of the aliphatic alcohol hexanediol, and more static, amyloid-like structures that are SDS insoluble and hexanediol resistant. Disaggregases, in particularly *hsp104*, were capable of re-dissolving heat or glucose-starvation induced granules, suggesting that the assembly of these kinds of structures may be dynamically regulated. The authors also utilize a somewhat novel genetically encoded particle assembled from a viral capsid protein to test the ability of different proteins to recruit P body and stress granule components. The following suggestions are offered for improving the manuscript:

1) The authors cleverly use hexanediol to disrupt P-bodies but not SGs. But they had to use two different strains to visualize P-bodies and SGs. This experiment would be more compelling if the authors could use a system that simultaneously labels P-bodies and SGs in the same cell and show that under the same conditions, hexanediol eliminates P-bodies but not SGs. For example, Buchan and colleagues recently used a yeast strain harboring Pab1-GFP and Edc3-mCherry as SG and P-body markers.

2) The amyloid-like state and P-body state Lsm4:Edc3 colocalizations shown in Figure 1 look somewhat different. Perfect co-localization for the amyloid-state ones vs. somewhat offset localization for the P-body ones. Is this observation reproducible? Can the authors speculate on the significance of this?

3) In Figure 6, the authors show that Nrp1 prion domain is necessary and sufficient to nucleate SGs. But are the Nrp1PLD foci really SGs? Do other SG markers also form and co-localize? The authors cleverly use the genetically encoded micro-particle to present the Nrp1 prion domain (Figure 6), which seems to nucleate the formation of SGs, using Nrp1-mCherry as a marker. Is this homo-typic interaction (full-length Nrp1 w/ prion domain of Nrp1) required for the subsequent recruitment of other SG proteins? In other words, could the authors repeat this experiment, but in an Nrp1∆ background and without using the Nrp1-mCherry marker. Would the Nrp1 prion domain still be sufficient to recruit other SG proteins via heterotypic prion domain associations? These experiments could be useful in supporting the hypothesis that SGs in yeast are formed by promiscuous interactions.

---

## [Author Response]

*1) The authors cleverly use hexanediol to disrupt P-bodies but not SGs. But they had to use two different strains to visualize P-bodies and SGs. This experiment would be more compelling if the authors could use a system that simultaneously labels P-bodies and SGs in the same cell and show that under the same conditions, hexanediol eliminates P-bodies but not SGs. For example, Buchan and colleagues recently used a yeast strain harboring Pab1-GFP and Edc3-mCherry as SG and P-body markers*.

We repeated the hexanediol experiment in a strain that simultaneously expressed GFP-tagged Nrp1 (SG marker) and mCherry-tagged Edc3 (PB marker) from their respective endogenous loci. The newly added Figure 4—figure supplement 1 shows that after glucose depletion, PB and SG formation is induced in this strain (“Before”). However, the addition of hexanediol to glucose-starved cells disrupts only PBs but not SGs (“After”).

*2) The amyloid-like state and P-body state Lsm4:Edc3 colocalizations shown in*
Figure 1
*look somewhat different. Perfect co-localization for the amyloid-state ones vs. somewhat offset localization for the P-body ones. Is this observation reproducible? Can the authors speculate on the significance of this*?

This experiment, as all the other experiments in the study, was performed with live cells to avoid artifacts that could be induced through chemical fixation. In living cells, however, some structures diffuse very rapidly. To image P bodies and other structures, we acquired z stacks of yeast cells. This required that we take multiple images of the same cell in different planes, often in two different channels (GFP and mCherry). Although we imaged the GFP and mCherry signals of each plane in quick sequential order, a movement of the P body during the acquisition often led to somewhat offset localization of the two signals. Thus, this does not reflect a true lack of colocalization, but is due to a technical problem. The same offset was not observed for amyloid-like assemblies, because they are usually larger and thus diffuse much more slowly.

*3) In*
Figure 6*, the authors show that Nrp1 prion domain is necessary and sufficient to nucleate SGs. But are the Nrp1PLD foci really SGs? Do other SG markers also form and co-localize? The authors cleverly use the genetically encoded micro-particle to present the Nrp1 prion domain (*Figure 6*), which seems to nucleate the formation of SGs, using Nrp1-mCherry as a marker. Is this homo-typic interaction (full-length Nrp1 w/ prion domain of Nrp1) required for the subsequent recruitment of other SG proteins? In other words, could the authors repeat this experiment, but in an Nrp1∆ background and without using the Nrp1-mCherry marker. Would the Nrp1 prion domain still be sufficient to recruit other SG proteins via heterotypic prion domain associations? These experiments could be useful in supporting the hypothesis that SGs in yeast are formed by promiscuous interactions*.

We thank the reviewers for this suggestion. To demonstrate that SGs can be nucleated through promiscuous heterotypic interactions, we repeated the experiment with plasmid-expressed fusions between µNS and Nrp1, Nrp1PLD or Nrp1∆PLD in a ∆*nrp1* background using Pub1-mCherry as SG marker, as suggested by the reviewers. Figure 6—figure supplement 12 shows that both the PLD and the RRM (∆PLD) domain of Nrp1 can nucleate Pub1-positive SGs under conditions of stress. No colocalization was observed under normal growth conditions. This shows that full-length Nrp1 is not required for SG formation and that heterotypic interactions are sufficient to nucleate SGs under stress conditions.